# Same Task, Different Circuits: Disentangling Modality-Specific Mechanisms in VLMs

Yaniv Nikankin[1]*    Dana Arad[1]    Yossi Gandelsman[2]    Yonatan Belinkov[1]

[1]Technion – Israel Institute of Technology    [2]UC Berkeley

## Abstract

Vision-Language models (VLMs) show impressive abilities to answer questions on visual inputs (e.g., counting objects in an image), yet demonstrate higher accuracies when performing an analogous task on text (e.g., counting words in a text). We investigate this accuracy gap by identifying and comparing the *circuits*—the task-specific computational sub-graphs—in different modalities. We show that while circuits are largely disjoint between modalities, they implement relatively similar functionalities: the differences lie primarily in processing modality-specific data positions (an image or a text sequence). Zooming in on the image data representations, we observe they become aligned with the higher-performing analogous textual representations only towards later layers, too late in processing to effectively influence subsequent positions. To overcome this, we patch the representations of visual data tokens from later layers back into earlier layers. In experiments with multiple tasks and models, this simple intervention closes a third of the performance gap between the modalities, on average. Our analysis sheds light on the multi-modal performance gap in VLMs and suggests a training-free approach for reducing it.[2]

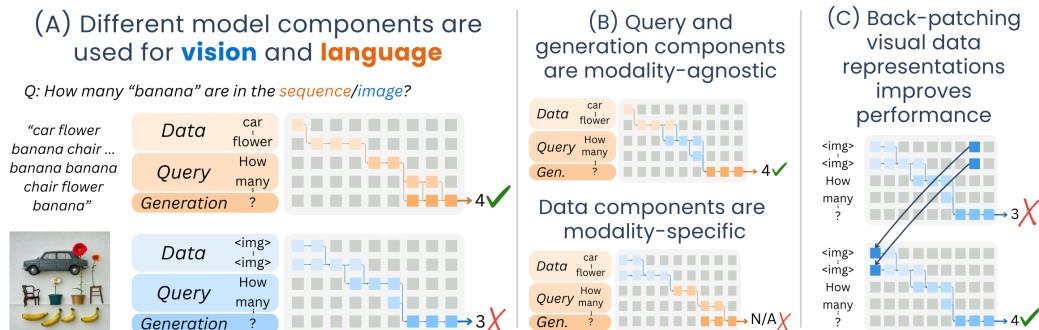

Figure 1: **Overview of our analysis.** (a) We find circuits for analogous vision and language tasks and show they are structurally disjoint—different model components are responsible for each modality. (b) Swapping sub-circuits across modalities (shown for the language circuit, but applies similarly to vision) reveals that query and generation components preserve performance when swapped between modalities, while swapping data components degrades performance. (c) To address the performance gap, we apply back-patching: re-injecting visual data activations from later layers into earlier ones. This makes textually-aligned representations from deeper layers available during visual prompt processing, enhancing performance on visual tasks.

---

*Correspondence to: yaniv.n@cs.technion.ac.il

[2]Code and data available at: https://github.com/technion-cs-nlp/vlm-circuits-analysis

# 1   Introduction

Recent years have seen major progress in building vision-and-language models (VLMs) that can handle both image and text inputs (Li et al., 2025). However, a significant performance gap persists between textual tasks and their visual counterparts (Fu et al., 2024; Wang et al., 2024a; Cohen et al., 2024). For example, models are considerably less accurate at counting objects in an image than at counting words in a list (Figure 1a, Appendix C), or at identifying a winner in a board game when it is supplied as an image rather than as a textual description (Fu et al., 2024). Curiously, this performance gap persists even in VLMs with a decoder that is jointly trained *from scratch* on both modalities (e.g. Team et al. (2024)). To address the performance gap between analogous image and text tasks in VLMs, one should first understand its source.

In this paper, we aim to explain the performance gap in VLMs between analogous visual and textual tasks from *structural* and *functional* perspectives. First, we create a dataset of five diverse tasks, each comprising pairs of analogous textual and visual prompt variants (Figure 2). Each prompt consists of data, query, and generation positions, that contain the subject of the prompt (an image or a short text), the task description, and the last token position, respectively. For example, the data for an object counting task can be a text listing objects ("car flower banana...") or an image with these objects, and the query for this task will ask "How many bananas are there?". We use this dataset to extract and examine the circuits—the computational sub-graphs—responsible for performing each of the variants. We measure the degree of *structural* overlap between these modality-specific circuits, and find that vision and language tasks are performed by relatively disjoint VLM components (Figure 1a), with an average of only 18% components shared between modalities.

To analyze the *functional* differences of intermediate sub-circuits, we swap them between the two modalities, and measure how interchangeable they are for task performance (Figure 1b). This reveals two main findings: first, despite being structurally disjoint, the sub-circuit that processes the query tokens, as well as the sub-circuit that generates the answer, are functionally equivalent between modalities—leading to similar performance when swapped. Second, there is a major functional difference in the processing of data tokens between modalities, resulting in much lower performance after swapping the relevant sub-circuit. Hence, we conclude that the difference in data processing is the leading cause behind the accuracy gap.

We utilize these findings to bridge the performance gap, without any additional training, by automatically intervening in the model's computation at test time. Specifically, we patch the visual token embeddings from later layers of the model—where these embeddings are more aligned to their textual counterparts—back to earlier layers (Figure 1c). This results in visual processing that is more aligned with analogous textual prompts, leading to higher accuracy on visual prompts. Our results suggest that understanding the differences in the mechanisms between modalities is attainable, and provides a path to improving visual processing in VLMs.

# 2   Preliminaries

In this section we present our main unit of analysis—model circuits. We discuss how circuits for specific tasks are discovered and evaluated. These discovery and evaluation techniques are used in Section 4 to identify the common and distinct sub-circuits for analogous tasks in different modalities.

## 2.1   Model Circuits

A circuit $\mathbf{c}$ is a minimal subset of interconnected model components that perform the computations required for a specific task or prompt (Elhage et al., 2021). In our analysis, we consider a circuit component to be either an entire attention head or an individual multilayer perceptron (MLP) neuron, at a specific output position. We follow the existing definition of MLP neurons (Geva et al., 2021; Nikankin et al., 2024), where the $n^{\text{th}}$ neuron in an MLP block is the combination of the $n^{\text{th}}$ row vector of the first MLP layer and the $n^{\text{th}}$ column vector of the second MLP layer. A full definition for MLP neurons and circuits is given in Appendix A.

## 2.2 Circuit Discovery

To identify circuits for both textual and visual task variants, we employ causal analysis techniques that score the importance of each component's activation (Vig et al., 2020). For each textual or visual task, we divide its prompts into two subsets: one for discovery and another for evaluation (See Appendix B.3). From the discovery subset, for each prompt $p$ with answer $r$, we select a random counterfactual prompt $p'$ that yields a different answer $r'$. Ideally, we would find the importance of a component $u$ for a prompt $p$ by intervening on its activation $a_{u,p}$ with the counterfactual activation $a_{u,p'}$, and calculate the difference in logits of $r$ and $r'$ post-intervention (Wang et al., 2022), marked $\text{LD}(r, r' | \text{do}(a_{u,p} = a_{u,p'}))$. However, this process becomes unfeasible due to the large amount of model components. Instead, we approximate this effect by applying attribution patching (Nanda, 2022) with integrated gradients (AP-IG; Sundararajan et al., 2017; Hanna et al., 2024):

$$\text{E}(u, p, p') = \left| (a_{u,p'} - a_{u,p}) \frac{1}{k} \sum_{i=1}^{k} \frac{\partial \text{LD}\left(r, r' | \text{do}\left(e = e' + \frac{i}{k}\left(e - e'\right)\right)\right)}{\partial a_{u,p}} \right| , \qquad (1)$$

where $e$ and $e'$ are the input embeddings for the prompts $p$ and $p'$, respectively, and $k = 5$ is the number of integration steps, following Hanna et al. (2024). This value is a first-order Taylor-series approximation to the logit difference, and can be calculated once, for all model components in parallel. The value is averaged across prompts and calculated per circuit component. A higher value indicates greater component importance for the analyzed task prompts. Given a score for each component, we construct a circuit $\mathbf{c}$ by selecting a specific percentage of the highest-scoring components.

## 2.3 Circuit Evaluation

We evaluate a circuit $\mathbf{c}$ by measuring its faithfulness (Wang et al., 2022)—the proportion of the full model's task performance explained by the circuit. For each prompt $p$ with answer $r$ in the evaluation subset, we randomly choose a counterfactual prompt $p'$ with answer $r'$ and compute the activations of the model for $p'$. We then pass $p$ through the model while ablating all non-circuit components using the pre-computed counterfactual activations, as per standard procedure (Mueller et al., 2025). The intervention's effect is quantified by the logit difference between correct and counterfactual answers, denoted $\text{LD}(r, r' | \text{do}(\forall u \notin \mathbf{c} : a_{u,p} = a_{u,p'}))$. This effect is normalized to calculate the faithfulness of circuit $\mathbf{c}$ on task T (Mueller et al., 2025):

$$\text{F}(\mathbf{c}, \text{T}) = \frac{1}{|\text{T}|} \sum_{(p,p',r,r') \in \text{T}} \frac{\text{LD}(r, r' | \text{do}(\forall u \notin \mathbf{c} : a_{u,p} = a_{u,p'})) - \text{LD}_{\mathbf{M}}(r, r')}{\text{LD}(r, r') - \text{LD}_{\mathbf{M}}(r, r')} , \qquad (2)$$

where $\text{LD}(r, r')$ is the logit difference without ablations and $\text{LD}_{\mathbf{M}}(r, r')$ is the logit difference when all components in model $\mathbf{M}$ are ablated. This normalization typically constrains faithfulness to a $[0.0, 1.0]$ range.

# 3 Experimental Settings

To investigate the accuracy gap between textual and visual prompts in VLMs, we construct a dataset of five question-answering tasks, seen in Figure 2. Each task consists of a query paired with data presented in one of two analogous formats: as either an image or a text. This analogous design ensures direct comparability—for every image-based prompt, we create an analogous text-based prompt that describes the same information. Our dataset contains the following tasks:

**Object counting:** The model is given a textual sequence of objects or an image containing these objects. The task is to count how many objects of a specific type appear in the data.

**Two-operand Arithmetic:** The model is given a two-operand arithmetic calculation (e.g., "56 + 14"), either as text or as a white-background image with the calculation written on it. The task is to complete the prompt with the correct answer.

**Spatial Ordering:** The model is presented with a scene with colored objects, either as an image from the CLEVR dataset (Johnson et al., 2017) or as a text describing the scene. The task is to identify the color of an object based on its position in the scene.

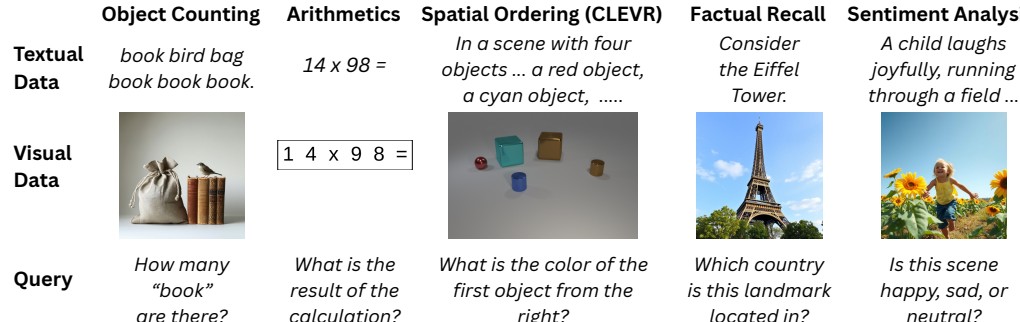

Figure 2: **Analogous Vision-Language Tasks.** We create a dataset of five question-answering tasks, each with a textual and visual variants. A task prompt is made up of a query (bottom row) asked either about an image (middle row) for the visual variant or on an analogous text (top row) for the textual variant.

**Factual Recall:** The model is presented with an entity—either a person or a landmark—either by name for the textual variant, or by image for the visual variant. The task is to complete a specific property of the entity. For example, when presented with a famous landmark, the task is to recognize which country is it located in.

**Sentiment Analysis:** The model is presented with a scene, either as an image or as the caption of an image. The task is to identify whether the sentiment in the scene is happy, sad, or neutral.

These were selected to span a diverse spectrum of tasks, and to incorporate both existing and generated data (i.e., factual recall uses real entity images while sentiment analysis and counting rely on generated images).

To perform circuit discovery on these tasks, we enforce two constraints on the prompts. First, we only use prompts where the answer is a single word. Second, for consistency within a task, we align all prompts to a positional template, such that each position in each prompt always contains the same type of token in it. (For instance, in factual recall, the first four tokens contain the entity name.) Prompts that do not align well to the template are filtered out. These constraints are standard in circuit discovery (Vig et al., 2020; Mueller et al., 2025). Prompt examples per task and further details on prompt generation are provided in Appendix B.2.

We analyze three transformer-based VLMs: Qwen2-7B-VL-Instruct (Wang et al., 2024b), Pixtral-12B (Agrawal et al., 2024), and Gemma-3-12B-it (Team et al., 2024). All these models use an "adapter" architecture (Liu et al., 2023), where the VLM comprises of a pre-trained image encoder that converts input image patches to a sequence of visual token embeddings, an optional adapter layer that projects each embedding to the language embedding space, and a language decoder. The decoder concatenates the visual embeddings with the rest of the textual prompt embeddings, and processes them to generate an answer. The models differ in their training schemes (e.g., Gemma-3's decoder is pre-trained from scratch on visual prompts, whereas Qwen2-7B-VL's decoder is initialized from the weights of a text-only LLM), training data, and additional techniques deployed to process visual information. This variety makes our findings robust in a wide array of settings.

To ensure we identify meaningful circuits within these models, we follow standard procedures (Mueller et al., 2025) and verify high VLM performance (substantially above-chance accuracy) on each task. We report accuracies in Appendix C and focus on the more common case where models achieve higher accuracy on textual variants compared to visual variants.

## 4 Cross-modality Circuit Analysis

To investigate the performance gap, we aim to pinpoint the differences between the circuits used for analogous visual and textual tasks. Namely, we analyze both the structural and functional intersection between these circuits. We start by identifying the circuits responsible for each combination of model, task, and modality.

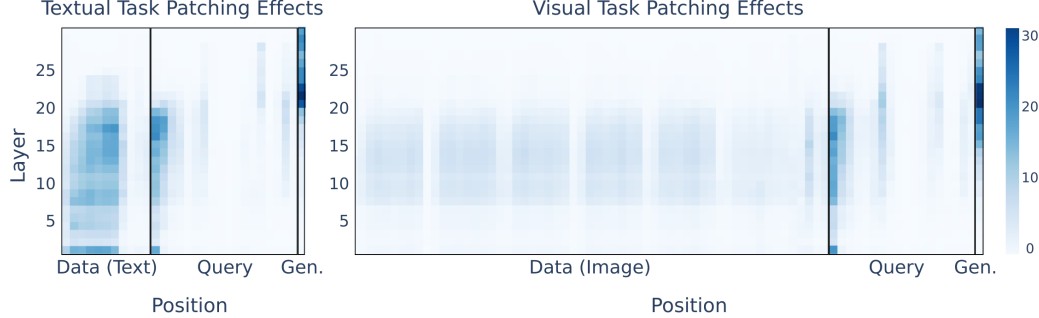

Figure 3: **Patching effects for Qwen2-7B-VL for the textual (left) and visual (right) counting task.** We sum the patching effect (described in Section 2.2) across all model components for a specific position and layer. This reveals different patterns of component importance by position, motivating the separation of each circuit to three sub-circuits—data, query and generation.

First, we use the methods described in Section 2 to find the patching effect of each model component, estimating their importance for the task. For instance, Figure 3 shows importance scores for the counting task in Qwen2-VL, summed per position and per layer (other models and tasks are shown in Appendix D). Next, we construct a circuit from the set of top-$p$[3] percent of components and measure its faithfulness, repeating this process for different circuit sizes. Following earlier work (Nikankin et al., 2024; Ameisen et al., 2025), a circuit is considered sufficient for a task if it is the minimal circuit that achieves a faithfulness of over 80%. As Figure 4 shows, the circuits we discover obtain high faithfulness results for relatively small circuit sizes, indicating that we find the most important components for the analyzed tasks, in *both* textual and visual modalities.[4] This allows to further focus on these circuits to analyze the structural (Section 4.1) and functional (Section 4.2) intersection between the modalities.

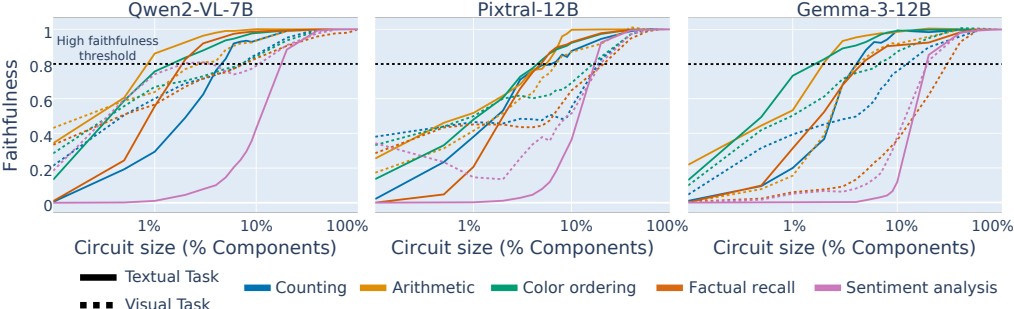

Figure 4: **Circuit faithfulness across models and tasks.** We measure the faithfulness at each circuit size, for each model, task and modality. The circuits we further analyze are the minimal circuits that achieve faithfulness of over 80%.

## 4.1 Vision and Language Circuits are Implemented by Different Components

Given the discovered sufficient circuits for each textual and visual task variants, we start by analyzing their structural intersection: how much component overlap is there between the two circuits?

Similarly to Kaduri et al. (2024), and motivated by Figure 3, we separate our prompts into three parts: data positions, query positions, and the answer generation position—the last token of the prompt. The positions are consistent within each modality, as detailed in Section 3, but can differ between

---

[3]$p \in \{0.001, 0.005\} \cup \{i \cdot 0.01 \mid i \in \mathbb{Z}, 1 \leq i \leq 100\}$

[4]On average, the textual task circuits we find include 7% of the components while the visual task circuits include 15% of the components. The larger amount of components necessary for visual tasks is expected since images have more tokens, requiring a larger number of position-aware components to cover them all.

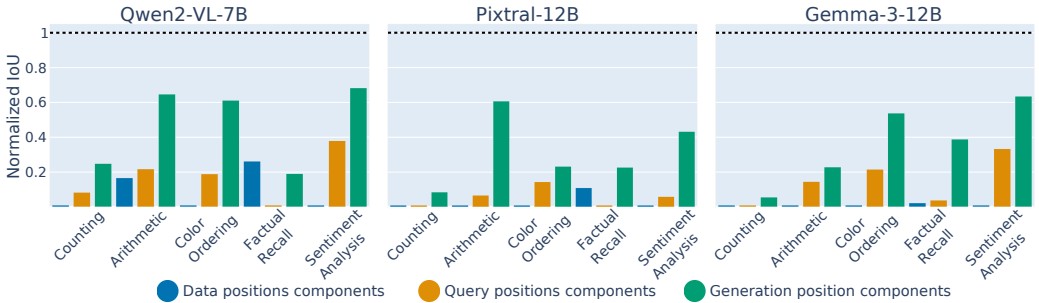

Figure 5: **Normalized IoU scores.** We measure the IoU between the component set of the textual and visual circuits for each model and task and normalize it using a random baseline. We find that across models and tasks, the intersection of components in data token positions (blue) is close to zero and the intersection of components in query token positions (orange) is very low. The intersection of components in the last token position (green) varies between tasks.

modalities. Since circuit components are position-specific, we have to map the positions between the two modalities in order to evaluate their overlap. Query positions and generation positions are easily mapped, since they consist of the same textual sequence for both textual and visual inputs (e.g., "How many 'banana' are in the image/sequence"?). A special case of alignment is in the data positions, where a single text position cannot be mapped to a single image patch. Thus, we do not consider data positions when calculating intersections for the data processing components (i.e., we only compare the layer and attention head index / MLP neuron index). This is formally defined in Appendix A.3.

Consequently, we split each circuit to three sub-circuits, split by positions: data ($\mathbf{c}^D$), query ($\mathbf{c}^Q$) and generation ($\mathbf{c}^G$) sub-circuits, which each include the circuit components in the corresponding positions, such that $\mathbf{c} = \mathbf{c}^D \cup \mathbf{c}^Q \cup \mathbf{c}^G$. A formal definition is given in Appendix A.2.

We use intersection over union (IoU) to quantify intersection between textual and visual circuits. Since the positional mapping between modalities is not one-to-one (due to visual prompts containing special tokens), we calculate IoU bidirectionally to ensure accurate comparison. First, we map the textual circuit's positions to the visual domain and measure its IoU with the visual circuit. Then, we perform the reverse operation, mapping the visual circuit's positions to the textual domain and measuring its IoU with the textual circuit. These two measurements are averaged to produce our intersection metric:

$$\text{IoU}\left(\mathbf{c}_L, \mathbf{c}_V\right) = \frac{1}{2}\left(\frac{|\mathbf{c}_{L\to V} \cap \mathbf{c}_V|}{|\mathbf{c}_{L\to V} \cup \mathbf{c}_V|} + \frac{|\mathbf{c}_L \cap \mathbf{c}_{V\to L}|}{|\mathbf{c}_L \cup \mathbf{c}_{V\to L}|}\right), \tag{3}$$

where $\mathbf{c}_L$, $\mathbf{c}_V$ are the textual and visual circuits, respectively. $\mathbf{c}_{L\to V}$ and $\mathbf{c}_{V\to L}$ are the textual and visual circuits with their positions mapped to the other modality, respectively (formally defined in Appendix A.3). Due to the significant numerical imbalance between MLP neurons and attention heads, we calculate the intersection in eq. (3) separately for MLP neurons and attention heads, and average the results.

To account for potential bias from varying circuit sizes (larger circuits may yield higher IoU scores), we establish a baseline using random circuits. For each model and task, we create two circuits $\mathbf{c}_{RL}$, $\mathbf{c}_{RV}$ from a set of random model components, each containing the same amount of components as the textual and visual circuits, respectively. We normalize the calculated IoU of the textual and visual circuits using the random baseline and a perfect IoU score of $1.0$:

$$\text{NIoU}\left(\mathbf{c}_L, \mathbf{c}_V\right) = \frac{\text{IoU}\left(\mathbf{c}_L, \mathbf{c}_V\right) - \text{IoU}\left(\mathbf{c}_{LV}, \mathbf{c}_{RV}\right)}{1.0 - \text{IoU}\left(\mathbf{c}_{LV}, \mathbf{c}_{RV}\right)}. \tag{4}$$

The normalized IoU results (Figure 5) reveal varying degrees of intersection across sub-circuits. In data positions, the modality-specific sub-circuits exhibit virtually no intersection across tasks and models—an expected finding given their specialized function in processing distinct data token distributions. In query positions, there is low overlap between modalities (12% on average), despite operating on identical input query tokens. The generation position sub-circuits show moderately higher intersection rates (38% on average), reflecting these components' shared role in promoting

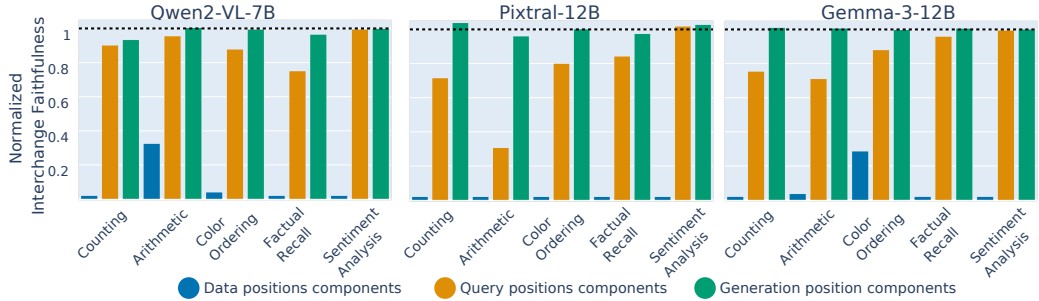

Figure 6: **Sub-circuit interchange faithfulness.** We measure the faithfulness of the textual and visual circuits while interchanging one of the sub-circuits with its parallel from the second modality. Across tasks, replacing the query (orange) or generation (green) sub-circuits between modalities has little effect on faithfulness, indicating shared functionality across modalities. In contrast, interchanging the data sub-circuit (blue) between modalities has a large negative effect on faithfulness.

consistent answers across modalities. Overall, these results indicate that while some components are important in both visual and textual tasks, the different modalities utilize relatively disjoint circuits.

### 4.2 Query Processing and Answer Generation are Functionally Equivalent; Data Processing is Modality-Specific

Despite being largely structurally disjoint, two distinct component sets may still implement similar functionality (Hanna et al., 2025). To measure the shared functionality of the textual and visual circuits, we build upon the partitioning of each circuit to three sub-circuits (Section 4.1). We investigate whether these sub-circuits are functionally interchangeable between the textual and visual circuits—for instance, can we replace $\mathbf{c}_L^Q$, the sub-circuit in the query positions of the language circuit, with $\mathbf{c}_V^Q$, the sub-circuit in the query positions of the visual circuit, while maintaining high circuit faithfulness? Such interchangeability would indicate shared functionality. Formally, we interchange the query sub-circuit between modalities and measure the average faithfulness (as in eq. (2)):

$$\frac{1}{2}\left(\mathrm{F}(\mathbf{c}_L^D \cup \mathbf{c}_{V \to L}^Q \cup \mathbf{c}_L^G, \mathrm{T}_L) + \mathrm{F}(\mathbf{c}_V^D \cup \mathbf{c}_{L \to V}^Q \cup \mathbf{c}_V^G, \mathrm{T}_V)\right) . \tag{5}$$

We apply this process to data and generation sub-circuits as well. When interchanging data sub-circuits, we ignore positional information (as in Section 4.1) and treat components as active across all data positions. As in eq. (4), we normalize these scores using a random baseline as a lower bound and the original faithfulness as an upper bound. The random baseline is measured by interchanging a sub-circuit with an identically-sized sub-circuit with random components, and the upper bound is the original faithfulness of the entire circuit ($\mathbf{c}_L$ or $\mathbf{c}_V$).

Our results (Figure 6) indicate that in the query and generation positions, textual and visual sub-circuits are interchangeable. In query positions, the visual sub-circuit can be replaced with a position-aligned textual sub-circuit (and vice versa), while maintaining high faithfulness to the entire model. This high interchange extends to the generation sub-circuits, showing a functional equivalence. Given the low structural overlap (12% and 38% on average in the query and generation positions, respectively, shown in Section 4.1), this implies that different components at these positions implement similar functionality, indicating a form of redundancy. A different pattern emerges when interchanging the sub-circuits at the data positions: across all tasks, the faithfulness drops to match the random baseline, indicating that the data sub-circuits $\mathbf{c}_L^D$ and $\mathbf{c}_V^D$ implement completely different functionality in textual and visual prompts. This indicates that the functional difference lies mainly in the components responsible for processing the data tokens, prior to being processed by the rest of the model.

## 5 Improving Visual Tasks Performance

Since textual and visual circuits functionally differ mainly in how they process analogous data tokens, we examine the representations at these positions to address the accuracy gap.

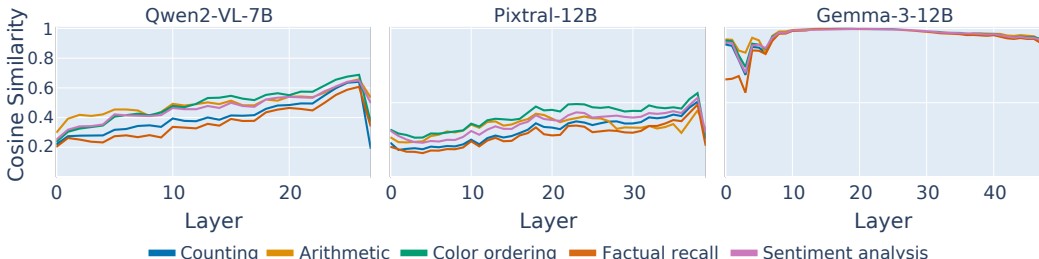

Figure 7: **Similarity of visual token activations to analogous textual token activations.** We compute cosine similarity between visual data activations and their analogous textual data activations. The maximum similarity for each visual token is selected and averaged across visual tokens. This similarity score increases deeper in the model, with varying rates across different models.

Recent work (Wu et al., 2024) has shown that as visual tokens move through a multi-modal model, they gradually align with the highest-matching textual token (e.g. visual tokens of a cat gradually align with the embedding of the token "cat"). We build upon this and measure the alignment of a whole image with its entire analogous textual prompt, across layers. We compute model activations for each analogous prompt pair. For each visual data token activation, we find the highest cosine similarity among *all* textual data token activations in the analogous prompt at the same layer. We average these to produce a cross-modality alignment score per layer.

We find that alignment between images and analogous texts gradually increases across layers (Figure 7). However, this increase is slow, with higher similarities only occurring later in the model, after information from data tokens moves to later positions (this information flow was shown in Kaduri et al. (2024), and is seen in Figure 3). We hypothesize that if these text-aligned representations were available earlier in the model's processing, it could improve performance on visual tasks.

To utilize the more textually-aligned representations of visual tokens, we employ back-patching (Biran et al., 2024; Lepori et al., 2024): the process of patching the late representations back to earlier layers (Figure 1c). For each visual prompt, we first perform a forward pass through the model and record activations $a^l$ for a window of consecutive layers centered at $l_{src}$, specifically $l \in \{l_{src} - i, \ldots, l_{src}, \ldots, l_{src} + i\}$ where $i \in \{0, 1, 2\}$ determines the window size (1, 3, or 5 layers). We then perform another forward pass, but this time replace the activations at a corresponding window centered at $l_{dst}$ (where $l_{dst} < l_{src}$) with our previously recorded activations from around $l_{src}$. This replacement is performed in parallel across all visual token positions. In a model with $L$ layers, we systematically explore this back-patching by varying the source layers $l_{src} \in \{\lfloor \frac{L}{2} \rfloor + 1, \ldots, L - 1\}$ and destination layers $l_{dst} \in \{0, 1, \ldots, \lfloor \frac{L}{2} \rfloor\}$, while maintaining the same window size for both source and destination, and choose the best performing $l_{src}, l_{dst}$ (reported in Appendix E.2).

To show statistical significance, we use bootstrap re-sampling with 1000 iterations and verify that the lower bound of the back-patched accuracy is higher than the baseline accuracy. We report the back-patched accuracies, the improvement compared to the baseline model accuracy, and standard deviations across resamples in Table 1.

Table 1: **Back-patching visual token embeddings increases accuracy across tasks.** We measure the improvement on visual tasks while back-patching visual token embeddings from later layers back to earlier layers. We report the back-patched accuracies and standard deviations, as well as the change from baseline model performance. Green marks statistically significant improvement. Back-patching shows an increase from baseline accuracies (reported in Appendix C) across most settings.

| Task | Qwen2-7B-VL | Pixtral-12B | Gemma-3-12B |
|---|---|---|---|
| Counting | 75.0% (+4.2% ± 1.9%) | 68.3% (+2.2% ± 1.1%) | 75.4% (+2.0% ± 2.1%) |
| Arithmetic | 76.9% (+9.4% ± 1.3%) | 29.4% (+6.7% ± 1.6%) | 94.2% (+6.7% ± 0.7%) |
| Color Ordering | 74.8% (+0.6% ± 0.6%) | 84.7% (+4.1% ± 1.4%) | 47.5% (+1.5% ± 0.9%) |
| Factual Recall | 69.7% (+1.6% ± 2.2%) | 49.1% (+4.1% ± 2.3%) | 68.6% (+2.1% ± 1.8%) |
| Sentiment Analysis | 94.8% (+2.2% ± 1.4%) | 87.4% (+17.7% ± 2.2%) | 98.0% (+5.4% ± 0.9%) |

Our results show that back-patching leads to an average absolute improvement of 4.6% on visual prompts, closing approximately 32% of the performance gap between visual and textual accuracy. Additional back-patching experiments are detailed in Appendix E.1 and Appendix E.4.

To confirm that these improvements stem from leveraging textually-aligned representations rather than mere additional computation, we conduct a control experiment and apply similar back-patching to textual prompt variants. This control yields smaller accuracy gains in roughly 79% of cases (further shown in Appendix E.3), supporting our hypothesis that improved visual performance is more due to the early transition of visual tokens to textually-aligned representations.

## 6 Related Work

**Interpretability of VLMs.**   Interpretability research has examined VLMs' internal mechanisms, investigating how they integrate visual and textual inputs to generate textual responses, starting from early visual question answering models (Agrawal et al., 2016) and sequence-to-sequence architectures (Chefer et al., 2021; Cao et al., 2020). Within this field, many studies have analyzed model weights and components and their effect on model outputs, both in discriminative VLMs (Gandelsman et al., 2025, 2024; Dravid et al., 2023; Goh et al., 2021) and generative VLMs (Kaduri et al., 2024; Zhang et al., 2024b; Jiang et al., 2024b; Neo et al., 2024; Luo et al., 2024; Yu & Ananiadou, 2024). Some studies have used causal analysis (Pearl, 2001), the study of causal mechanisms in computational graphs, in VLMs to identify key components in varied tasks (Li et al., 2022; Basu et al., 2024; Golovanevsky et al., 2024). Building on these, we identify model circuits for a range of analogous tasks, and measure their structural and functional overlap. In LLMs, recent work shows high circuit overlap for similar tasks (Merullo et al., 2024; Zhang et al., 2024a; Mondorf et al., 2024; Hanna et al., 2025). We demonstrate this phenomenon occurs in VLMs at a different granularity by investigating position-based sub-circuits, and propose a simple inference-time method to improve visual accuracy.

**Multi-modal representations.**   Previous work has shown that representations from models trained on different modalities, like vision and language, can be aligned using transformations to build multi-modal models (Merullo et al., 2022; Koh et al., 2023). The VLMs we investigate implement similar approaches. However, since models perform differently across modalities (Fu et al., 2024; Cohen et al., 2024; Wang et al., 2024a; Ventura et al., 2024), it raises the question: *how* do these modalities align, and *where* do they differ? Recent studies have explored multi-modal alignment in different ways: showing a gap between textual and visual representations (Liang et al., 2022; Jiang et al., 2024a), investigating how in-context tasks are represented (Luo et al., 2024) and how semantically similar tokens are aligned (Wu et al., 2024). While these findings help us understand modality alignment better, they do not explain why accuracy varies across modalities. Building upon previous research that investigated a representational gap in contrastive VLMs (Liang et al., 2022) and its correlation with accuracy (Schrodi et al., 2024), we propose a way to modify under-performing visual inputs so they can better match textual representations and improve model accuracy.

## 7 Conclusions

Our research explores why VLMs perform differently when processing the same tasks on textual or visual data. We create a dataset of five analogous tasks in both modalities to investigate this accuracy gap. We discover that even models trained simultaneously on both modalities develop separate circuits for each. Despite being separate, the circuits are functionally equivalent for the most part—when processing query tokens and generating the answer—but functionally differ when processing data tokens. Using these findings, we utilize a simple test-time method, back-patching, to increase VLM accuracy on visual prompts and reduce the accuracy gap between modalities.

Our results demonstrate that visual tokens benefit from textual alignment through further model processing. This points to the benefit of flexible processing for different tokens in VLMs. Relatedly, Geiping et al. (2025) presented a new paradigm of test-time compute scaling, where models use a varying amount of layers per prompt. Future work can extend this and explore the benefit of varied computational resources *per token*, particularly for visual inputs which may require more processing than text.

# 8  Limitations

We focused on question-answering tasks involving single images, where the image appears prior to the query. Further work could explore multi-image scenarios, different prompt order, or more complex and visually-inclined reasoning tasks, where the model is expected to perform better on visual representations (e.g., navigating through a visual environment). We limited our scope to simpler tasks with single-token completions due to two key constraints. First, current open models struggle with more complex visual reasoning, making mechanism analysis less meaningful in these cases, and second, current circuit analysis methods can only be applied to single-token completions.

Our experiments show that patching visual token representations from later to early layers closes 32% of the performance gap between analogous visual and textual prompts. Yet, the causes for the remaining gap remain unclear, suggesting potential for further visual processing improvements.

Finally, we focus our analysis on adapter-based VLMs, as this is the current state-of-the-art approach for aligning vision and language. Our findings may not immediately apply to models utilizing different fusing approaches, which we leave for future work.

## Acknowledgments

We thank Tal Haklay for her feedback on this project, as well as Simon Schrodi, Jiahai Feng, and Tomer Ashuach for their helpful comments on drafts of this paper. We thank the anonymous reviewers of this paper for providing useful feedback. This research was supported by an Azrieli Foundation Early Career Faculty Fellowship and Open Philanthropy. DA is supported by the Ariane de Rothschild Women Doctoral Program. YG is supported by the Google Fellowship. This research was funded by the European Union (ERC, Control-LM, 101165402). Views and opinions expressed are however those of the author(s) only and do not necessarily reflect those of the European Union or the European Research Council Executive Agency. Neither the European Union nor the granting authority can be held responsible for them.

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

# A Formal circuit definitions

## A.1 Definition of MLP Neurons

Our definition of MLP neurons as circuit components follows existing literature (Nikankin et al., 2024), and is provided for completeness. In the VLMs we analyze, the MLP layer of a transformer block is implemented by a Gated MLP (Liu et al., 2021). This MLP at layer $l$ can be described by the following equations:

$$\mathbf{h}^l_{post} = \sigma(\mathbf{h}^l_{in}\mathbf{W}^{l\top}_{gate}) \circ (\mathbf{h}^l_{in}\mathbf{W}^{l\top}_{in}) , \tag{6}$$

$$\mathbf{h}^l_{out} = \mathbf{h}^l_{post}\mathbf{W}^l_{out} , \tag{7}$$

where $\mathbf{h}^l_{in}, \mathbf{h}^l_{out} \in \mathbb{R}^d, \mathbf{h}^l_{post} \in \mathbb{R}^{d_{mlp}}$ are the MLP input, MLP output and post-non-linearity representations of the MLP block, respectively. $\mathbf{W}^l_{in}, \mathbf{W}^l_{out} \in \mathbb{R}^{d_{mlp}\times d}, \mathbf{W}^l_{gate} \in \mathbb{R}^{d_{mlp}\times d}$ are weight matrices, and $\sigma$ is a non-linearity function, $\circ$ is Hadamard product, and biases are omitted.

In this formulation, we define the activation of the $n^{th}$ MLP neuron in layer $l$ as the $n^{th}$ scalar value in the intermediate representation $\mathbf{h}^l_{post}$. This scalar is multiplied with the $n^{th}$ row of $\mathbf{W}^l_{out}$, to produce the neuron's contribution to the output of the entire MLP block.

## A.2 Definition of circuits

A circuit $\mathbf{c}$ is formally defined as a set of components:

$$\mathbf{c} = \{(l, p, i) \mid l \in \mathbb{N}, p \in \mathbb{N}, i \in \mathbb{N}\}$$

where each component is represented by a 3-tuple $(l, p, i)$ denoting the layer, position, and index respectively. Each component represents either an attention head (where $i$ denotes the attention head index within layer $l$) or an MLP neuron (where $i$ denotes the neuron index within layer $l$, as defined in Appendix A.1).

When decomposing a circuit into its data ($\mathbf{c}^D$), query ($\mathbf{c}^Q$), and generation ($\mathbf{c}^G$) sub-circuits, the circuit components are partitioned according to their position indices, such that $\mathbf{c} = \mathbf{c}^D \cup \mathbf{c}^Q \cup \mathbf{c}^G$. Formally, this decomposition is defined as:

$$\mathbf{c}^D = \{(l, p, i) \in \mathbf{c} \mid p \in P_D\}$$

$$\mathbf{c}^Q = \{(l, p, i) \in \mathbf{c} \mid p \in P_Q\}$$

$$\mathbf{c}^G = \{(l, p, i) \in \mathbf{c} \mid p \in P_G\}$$

where $P_D$, $P_Q$, and $P_G$ are disjoint sets of position indices corresponding to the data, query, and generation regions respectively. In our case, the generation region includes only the last input position.

## A.3 Definition of circuit component mapping

A positional mapping is a function $M : \mathbb{N} \to \mathcal{P}(\mathbb{N})$ that maps each position in a textual task prompt to one or more positions in a visual task prompt containing equivalent information, and vice versa to map visual positions to textual positions (marked $M^{-1}$). We manually define the positional mapping for each task and model combination. For an individual circuit component $(l, p, i)$, the mapping transforms it to a set of components in the target modality:

$$M\left((l, p, i)\right) = \{(l, p', i) \mid p' \in M(p)\} .$$

For example, in the object counting task with Qwen2-7B-VL, the mapping function $M$ maps textual position 23 to visual position 96, where both contain the first token of the query ("How"). The complete positional mappings for all tasks and models are provided in the code accompanying the paper.

To compute our similarity metrics (Equation (4), Equation (5)) over query and generation sub-circuits, we map a sub-circuit from one modality to another (e.g. $\mathbf{c}^Q_{L \to V}$) by applying the mapping function to

each component individually. Formally, for a query ($\mathbf{c}^Q$) or generation sub-circuit ($\mathbf{c}^G$), we obtain the mapped sub-circuit:

$$\mathbf{c}_{L \to V} = M(\mathbf{c}) = \bigcup_{(l,p,i) \in \mathbf{c}} M((l, p, i)) .$$

The treatment of data sub-circuits differs because a positional correspondence does not always exist between modalities in the data positions. For instance, in the sentiment analysis task, general words in the textual sequence may not map to specific visual tokens, but rather convey an overall sentiment or atmosphere that is distributed across the entire visual input. Therefore, when computing similarity metrics for data sub-circuits $\mathbf{c}^D$, we discard positional information and represent components as 2-tuples $(l, i)$.

# B  Experimental and Technical Details

## B.1  Prompt examples

Table 2 presents prompt examples for each task in both input modalities.

Table 2: Textual and Visual Prompt Examples per task

| Task Name | Textual Prompt Example | Visual Prompt Example |
|---|---|---|
| Object Counting | "Sequence: book tree book cup cup ball tree. How many "tree" are in the sequence? Answer in a single number" | "How many "tree" are in the image? Answer in a single number." |
| Arithmetic | "Question: 10*48. What is the result of the given arithmetic calculation? Answer in a single number" | 10 x 46 = "What is the result of the given arithmetic calculation? Answer in a single number." |
| Spatial Ordering | "In a scene with four objects arranged horizontally, there is a green object, a yellow object, a yellow object and a cyan object. What is the color of the third object from the left? Answer in a single word." |  "What is the color of the third object from the left? Answer in a single word." |
| Factual Recall | "Consider Dennis Rodman. What sport does this athlete play? Answer in a single word." |  "What sport does this athlete play? Answer in a single word." |
| Sentiment Analysis | " "A child runs through a field of daisies, laughter echoing in the sunshine, feeling the ...". Is this scene happy, sad, or neutral? Answer in a single word." | "Is this scene happy, sad, or neutral? Answer in a single word." |

The amount of prompts per task is shown in Table 3. The amount of prompts for the same task and modality varies between models due to tokenization considerations (e.g. all object colors in the spatial ordering task must be tokenized to a single token for positional alignment purposes—prompts that contain multi-token object colors are filtered out).

The full list of prompts will be published.

Table 3: **Prompt amounts for each model, task and modality.**

| Task | Qwen2-7B-VL | | Pixtral-12B | | Gemma-3-12B | |
|------|------|------|------|------|------|------|
| | L | V | L | V | L | V |
| **Counting** | 1524 | 383 | 1524 | 334 | 1524 | 383 |
| **Arithmetic** | 1000 | 1000 | 1000 | 1000 | 1000 | 1000 |
| **Spatial Ordering** | 1865 | 1925 | 472 | 497 | 1865 | 1925 |
| **Factual Recall** | 416 | 1265 | 454 | 1265 | 453 | 1265 |
| **Sentiment Analysis** | 232 | 245 | 237 | 245 | 222 | 245 |

## B.2 Tasks Generation Details

In this section we describe in detail the prompt generation process for each task and modality in our dataset.

**Object Counting:** We use a base list of 30 possible object types (e.g. "banana", "fork", "book"). Each textual prompt contains seven randomly sampled objects drawn from up to four different object types—a configuration chosen to balance task difficulty while maintaining high VLM accuracy. For the visual prompts, we use SD3-XL (Podell et al., 2023) to generate images containing the same amount of objects described in the text. Due to limitations in image generation models' ability to produce exact object counts, we manually verify each image, and filter out images that don't match the required object counts.

**Arithmetic:** For the textual variant, each prompt consists of two two-digit operands, ensuring positional alignment between different prompts (e.g., the second position always includes the singles digit of the first operand). This consistency is important because all analyzed models tokenize numbers into separate digits. For the visual variant, we create white-background images (75×338 resolution) displaying the identical arithmetic calculations in large black font centered in the image, maintaining a simple presentation that focuses entirely on the calculation itself.

**Spatial Ordering:** For the visual variant, the images are taken from the CLEVR dataset (Johnson et al., 2017). Each image depicts a scene with four colored objects. We select this specific object count as it provides an optimal difficulty tradeoff—scenes with three objects prove too easy for models, while scenes with five objects result in significantly lower model accuracy. The textual variant is generated out of these scenes. Each textual prompt depicts a CLEVR scene with four colored objects, mentioning only their colors without shapes, to maintain positional alignment (as different shapes (e.g. "sphere") get tokenized into varying numbers of tokens).

**Sentiment Analysis:** We use ChatGPT (Hurst et al., 2024) to generate 120 scene descriptions for each sentiment—happy, sad, or neutral, and manually drop scene descriptions that don't convey the target sentiment. To maintain positional alignment in the textual variant, we truncate each description to exactly 20 tokens and pad the description with "...". Scene descriptions shorter than 20 tokens are filtered out. We use a scene token length of 20 since most scenes are longer than it, and we found it to convey the sentiment of the scene well enough in all cases. For the corresponding visual prompts, we use Flux1-Schnell (Labs, 2024) to generate images based on these scene descriptions, conducting additional manual filtering to exclude any images that don't adequately match the described scene or fail to convey the intended sentiment.

**Factual Recall:** We use three prompt templates from Hernandez et al. (2023):

1. "Consider [X]. What sport does this athlete play? Answer in a single word."
2. "Consider [X]. Which country is this landmark in? Answer in a single word."
3. "Consider [X]. What instrument does this person play? Answer in a single word."

For the textual variant, we select entities (to replace the [X] in the prompt, e.g. "Michael Jordan") that, when combined with the prefix "Consider [X]", result in exactly 5 tokens to ensure positional alignment. We use this number as it results in the highest number of possible entities in the given

dataset. For the visual variant, we collect entity images with public licenses from Wikimedia Commons. We rank 10 potential images per entity using CLIP (Radford et al., 2021) scores and select the best match for each entity. We resize and center-crop each image (to 256x256 resolution). We manually review the images to remove any that contain "shortcut" hints (e.g., any image of a basketball player holding a basketball), to ensure models rely on factual recall rather than visual cues. In both variants, the counterfactual prompts for each template are sampled only from within the same template.

## B.3 Circuit Discovery and Evaluation

For our circuit discovery experiments (detailed in Section 2 and Section 4), we allocate each modality-specific task dataset with a 75/25 split: 75% of the prompts for discovery and 25% of the prompts for faithfulness evaluation. We divide prompts between the discovery and evaluation subsets such that all possible answers are distributed in the same ratio between the subsets.

When answers span multiple tokens (e.g. in the arithmetic task, where answers can be comprised of multiple digits), we measure the patching effect (Equation (1)) solely on the first token. We verify that the answers for each prompt and counterfactual prompt ($r$ and $r'$) differ in the analyzed first token.

Our circuit discovery experiments use our fork of the TransformerLens library (Nanda & Bloom, 2022), in which we implement patching code for VLMs. This fork is available in our code release.

## B.4 Compute Resources

Our experiments were conducted on an NVIDIA L40 node equipped with 8 GPUs, each containing 48GB of memory. Peak memory consumption occurred during circuit discovery operations on the Gemma-3-12B-Instruct model, that required the parallel use of 4 GPUs. The complete experimental suite, including studies not featured in the final paper, consumed roughly 200–300 GPU hours.

## C Task Accuracies

In Table 4 we report the accuracies of models on each of the tasks. We observe that across most tasks, VLMs achieve higher accuracy in the textual variant than the visual variant, motivating our analysis into this performance gap. This is compatible with earlier results on closed-source VLMs (Fu et al., 2024).

Table 4: **Model Accuracies for textual and visual task variants.**

| Task | Qwen2-7B-VL | | Pixtral-12B | | Gemma-3-12B | |
| | L | V | L | V | L | V |
|---|---|---|---|---|---|---|
| **Counting** | 79.3% | 70.8% | 49.3% | 66.1% | 89.3% | 73.4% |
| **Arithmetic** | 99.2% | 67.5% | 25.7% | 22.7% | 99.0% | 87.5% |
| **Color Ordering** | 86.8% | 74.2% | 77.4% | 80.6% | 76.1% | 46.0% |
| **Factual Recall** | 73.5% | 68.1% | 83.4% | 45.0% | 80.7% | 66.5% |
| **Sentiment Analysis** | 97.4% | 92.6% | 98.3% | 69.7% | 98.7% | 92.6% |

## D Additional Circuit Discovery Results

In Table 5 we report the circuit size, in a percentage of the model's component count. As described in Section 4, the circuits are chosen to be the minimum-sized circuit with a faithfulness of over 80%. The faithfulness achieved by each circuit on each task and modality is reported in Table 6.

Furthermore, in Figure 10, Figure 11, Figure 12, Figure 13, Figure 14 we present the patching importance scores for each model and task, summed per position and layer. These figures are complementary to Figure 3.

Table 5: **Circuit sizes, in percentage of components out of the model's whole component set.**

| Task | Qwen2-7B-VL | | Pixtral-12B | | Gemma-3-12B | |
|---|---|---|---|---|---|---|
| | L | V | L | V | L | V |
| **Counting** | 5% | 8% | 7% | 20% | 5% | 20% |
| **Arithmetic** | 1% | 3% | 6% | 7% | 2% | 5% |
| **Color Ordering** | 2% | 8% | 5% | 20% | 2% | 7% |
| **Factual Recall** | 2% | 8% | 5% | 20% | 5% | 20% |
| **Sentiment Analysis** | 2% | 10% | 20% | 20% | 10% | 20% |

Table 6: **Circuit faithfulness scores for textual and visual task variants.**

| Task | Qwen2-7B-VL | | Pixtral-12B | | Gemma-3-12B | |
|---|---|---|---|---|---|---|
| | L | V | L | V | L | V |
| **Counting** | 82.9% | 82.0% | 81.4% | 82.7% | 88.0% | 89.7% |
| **Arithmetic** | 86.0% | 81.9% | 82.6% | 80.3% | 80.9% | 83.9% |
| **Color Ordering** | 83.6% | 81.2% | 82.4% | 84.1% | 82.8% | 81.3% |
| **Factual Recall** | 82.6% | 82.7% | 80.7% | 94.6% | 82.3% | 92.8% |
| **Sentiment Analysis** | 88.2% | 80.4% | 92.4% | 80.7% | 85.3% | 82.1% |

# E  Back-Patching Ablations and Extended Results

## E.1  Iterative Back-patching

Extending the back-patching experiments in Section 5, we explore if additional processing of visual data tokens yields further improvements. To do that, we perform back-patching iteratively (back-patching more than once). After identifying the optimal layers $l_{src}, l_{dst}$ and layer window size for each model and task, we perform back-patching iteratively. Namely, after each patching of the window centered at $l_{dst}$, we continue the forward pass to calculate the window centered at $l_{src}$, and re-patch the window centered at $l_{dst}$. This process is repeated up to 10 times. As shown in Figure 8, performance degrades after the first back-patching iteration across all but one task and model pairs, with each subsequent iteration further reducing model accuracy. We attribute this decline to representations becoming increasingly out-of-distribution with each iteration, making them progressively less compatible with the model's learned parameters.

## E.2  Best Layers For Back-Patching

For each model and task, Table 7 shows the values of $l_{src}, l_{dst}$ and the layer window size that lead to the highest back-patching results, reported in Table 1.

These optimal layers match our hypothesis across models: replacing visual data token representations at model layers when they are less textually-aligned, with more textually-aligned representations,

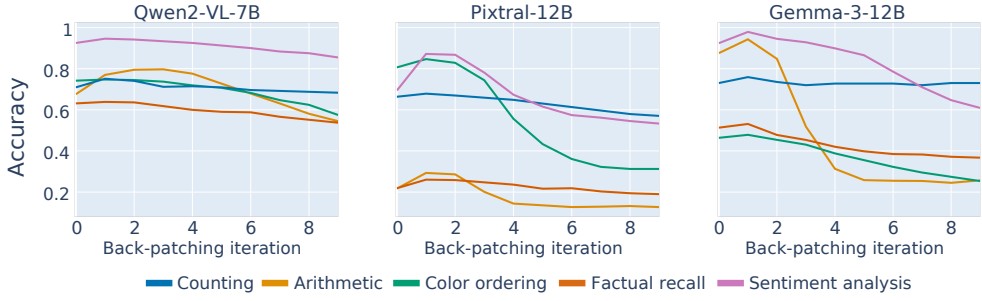

Figure 8: **Iterative back-patching results.** Applying back-patching several times in the highest-performing settings leads to a decrease in accuracy after the first back-patching application.

Table 7: **Best back-patching settings.** Each cell presents the $l_{src}, l_{dst}$ and layer window size (in parenthesis) that leads to the highest back-patching accuracy.

| Task | Qwen2-7B-VL | Pixtral-12B | Gemma-3-12B |
|---|---|---|---|
| **Counting** | $20 \rightarrow 15 \;\; (3)$ | $28 \rightarrow 23 \;\; (5)$ | $36 \rightarrow 5 \;\; (3)$ |
| **Arithmetic** | $11 \rightarrow 7 \;\; (3)$ | $37 \rightarrow 10 \;\; (5)$ | $32 \rightarrow 5 \;\; (1)$ |
| **Color Ordering** | $18 \rightarrow 15 \;\; (1)$ | $32 \rightarrow 22 \;\; (5)$ | $34 \rightarrow 24 \;\; (1)$ |
| **Factual Recall** | $11 \rightarrow 6 \;\; (5)$ | $32 \rightarrow 18 \;\; (5)$ | $34 \rightarrow 20 \;\; (5)$ |
| **Sentiment Analysis** | $15 \rightarrow 5 \;\; (5)$ | $37 \rightarrow 19 \;\; (5)$ | $23 \rightarrow 3 \;\; (5)$ |

yields the highest accuracy improvements. This pattern persists even in Gemma-3-12B, where visual data tokens demonstrate high alignment with textual analogs throughout most of the model (as shown in Figure 7): tasks showing the greatest accuracy gains from back-patching (counting, arithmetic, and sentiment analysis, as detailed in Table 1) benefit from patching in the earliest layers where the alignment of tokens with textual analogs is relatively low.

### E.3 Back-Patching Control Experiments

This section presents our back-patching control experiment results across models and tasks. The accuracy increases from back-patching visual data tokens (reported in Table 1) can either stem from earlier textually-aligned representations or from additional compute. To isolate the effect of adding computational layers, we conduct the same back-patching analysis on analogous textual prompts. In this control experiment, the alignment with textual tokens is inherent, so any accuracy increase must stem from added compute. Across each back-patching setting—model, task and layer window size—we check for how many values of $l_{src}$ and $l_{dst}$ is the accuracy increase caused by back-patching larger than the control accuracy increase. Our findings (Figure 9) show that in most settings, visual back-patching yields greater accuracy improvements than the control for a majority of layer values. A notable exception occurs in factual recall tasks for Qwen2-VL-7B and Gemma-3-12B, where back-patching shows minimal accuracy gains and is outperformed by the control in over half the cases. These results support our hypothesis that improved visual performance stems primarily from the early transition of visual tokens into textually-aligned representations, rather than from increased computational depth.

### E.4 Back-Patching in General VQA

To demonstrate that back-patching generalizes beyond our constructed dataset in broader visual question-answering, we evaluate its impact on model accuracy using the VQAv2 (Goyal et al., 2017) and RealWorldQA (xAI, 2024) datasets, two common VQA benchmarks. We measure the accuracy of each VLM on each dataset without back-patching as a baseline, and measure it again when applying back-patching (as described in Section 5). We present the results in Table 8. For VQAv2, we measure the accuracy of each analyzed VLM on 3000 randomly sampled visual prompts (ensuring no image repetition). For RealWorldQA, we measure the accuracy on the entire dataset of 765 prompts. We limit the analysis to this amount of prompts due to computational requirements of exploring different back-patching settings. For both datasets, we also evaluate the mean and standard deviation of the accuracy by bootstrapping for 1000 iterations with a full sample size. In both datasets, following our experimental setup for other tasks (Appendix B.3, we sample the model for a single forward pass, forcing an immediate answer. While this leads to generally lower results on the benchmarks, this setup is necessary due to the need to search for the best-performing back-patching settings, and ensures consistency across our experimental framework.

The results confirm the average improvements observed in Section 5. Since the visual prompts in VQAv2 and RealWorldQA do not have textual analogs, we do not perform the control experiment in this setting.

### E.5 Back-patching across model scales

To evaluate if and how back-patching visual data representations affects accuracy across model scales, we perform back-patching on 3 scales of the PaliGemma2 VLM suite (Steiner et al., 2024): 3B, 10B

Table 8: **Back-patching results on general VQA benchmarks.** Back-patching leads to a significant increase in accuracy in general VQA benchmarks as well.

| Dataset | Setting | Qwen2-7B-VL | Pixtral-12B | Gemma3-12B |
|---------|---------|-------------|-------------|------------|
| **VQAv2** | Baseline | 65.0% | 62.2% | 65.7% |
| | Back-patching | $68.2\% \pm 0.9\%$ | $66.2\% \pm 1.1\%$ | $70.3\% \pm 1.3\%$ |
| **RealWorld** | Baseline | 58.8% | 58.3% | 59.1% |
| | Back-patching | $61.6\% \pm 1.0\%$ | $60.1\% \pm 1.2\%$ | $60.0\% \pm 0.8\%$ |

and 28B parameters. We evaluate the models on our dataset of five tasks, and measure the mean accuracy and it's standard deviation by bootstrapping for 1000 iterations with full sample size. The baseline accuracies are presented in Table 9. The back-patched accuracies are reported in Table 10 and show that across most model scale and task pairs, back-patching increases model accuracy, by aligning visual token representations with their analogous textual representations earlier in the model.

Table 9: **Baseline PaliGemma2 accuracies.**

| Model Size | Counting | Arithmetic | Color Ordering | Factual Recall | Sentiment Analysis |
|------------|----------|------------|----------------|----------------|--------------------|
| **3B** | 64.0% | 0.7% | 15.4% | 32.7% | 48.7% |
| **10B** | 66.0% | 8.7% | 14.0% | 38.7% | 56.7% |
| **28B** | 22.0% | 6.7% | 12.0% | 30.7% | 10.7% |

Table 10: **Back-patching accuracy increase across model scales.** We evaluate how back-patching affects the accuracy of the PaliGemma2 VLM suite in three model sizes.

| Model Size | Counting | Arithmetic | Color Ordering | Factual Recall | Sentiment Analysis |
|------------|----------|------------|----------------|----------------|--------------------|
| **3B** | $+1.3\% \pm 2.9\%$ | $+31.6\% \pm 3.8\%$ | $+8.1\% \pm 3.4\%$ | $+6.5\% \pm 3.9\%$ | $+7.6\% \pm 3.9\%$ |
| **10B** | $+3.4\% \pm 3.1\%$ | $+28.1\% \pm 4.1\%$ | $+11.9\% \pm 3.4\%$ | $+17.5\% \pm 4.0\%$ | $+16.1\% \pm 3.6\%$ |
| **28B** | $+4.2\% \pm 3.6\%$ | $+24.1\% \pm 3.8\%$ | $+3.5\% \pm 2.9\%$ | $+5.3\% \pm 4.0\%$ | $+15.3\% \pm 3.6\%$ |

A note-worthy finding in the results is that the accuracy increase caused by back-patching is the largest in the 10B model size. We hypothesize this is caused by the overall lower performance of the 3B and the 28B model size. According to Steiner et al. (2024), the lower performance at the 28B size likely happens because this model scale is trained from scratch, as opposed to the smaller scales that are a result of model distillation.

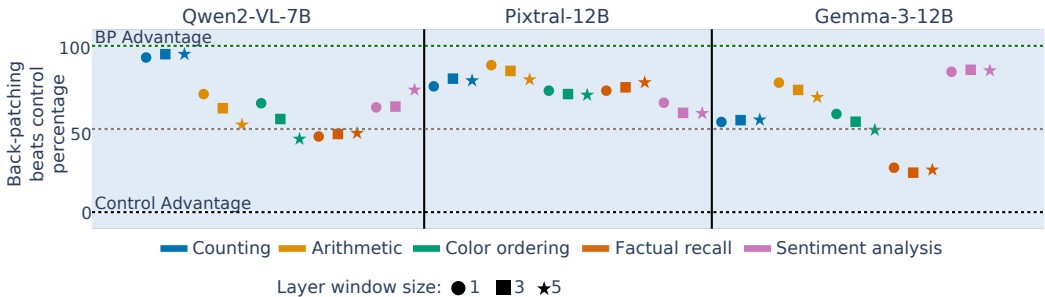

Figure 9: **Back-patching vs control results.** We measure the percentage of $l_{src}, l_{dst}$ values for which the control accuracy increase is lower than the back-patching accuracy increase. In most model, task and layer window size combinations, back-patching shows higher accuracy increase (above the 50% line) compared to the control results.

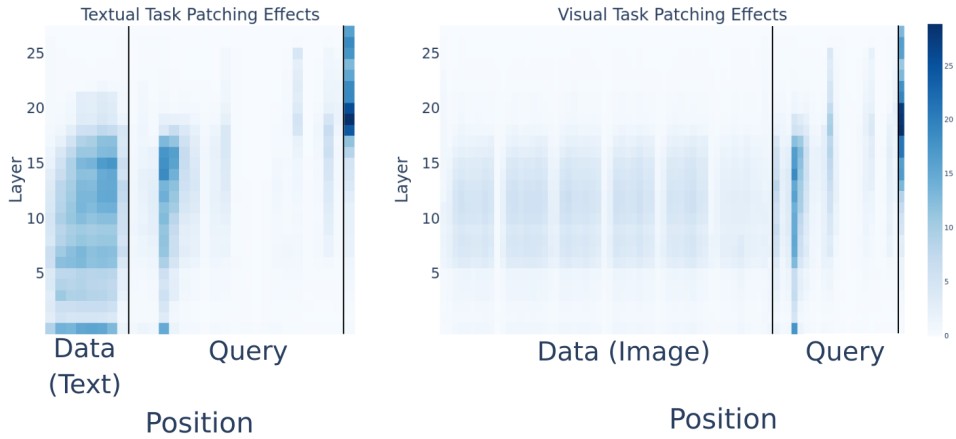

(a) **Patching effects for Qwen-7B-VL for the textual (left) and visual (right) counting task.**

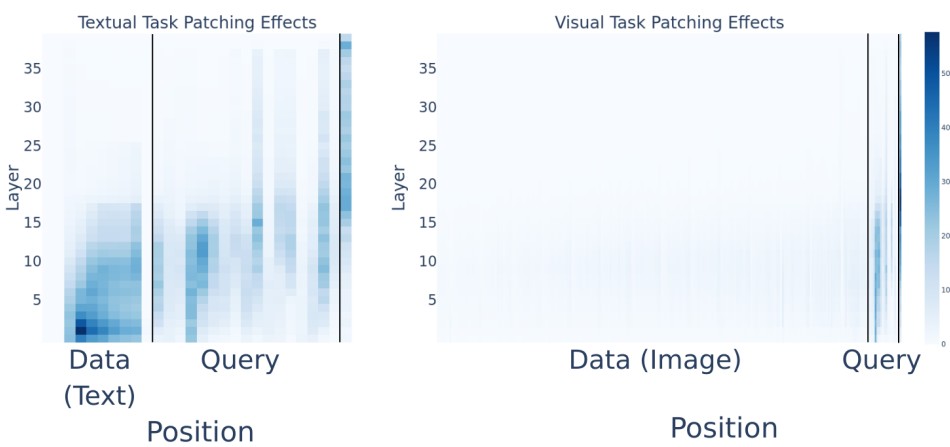

(b) **Patching effects for Pixtral-12B for the textual (left) and visual (right) counting task.**

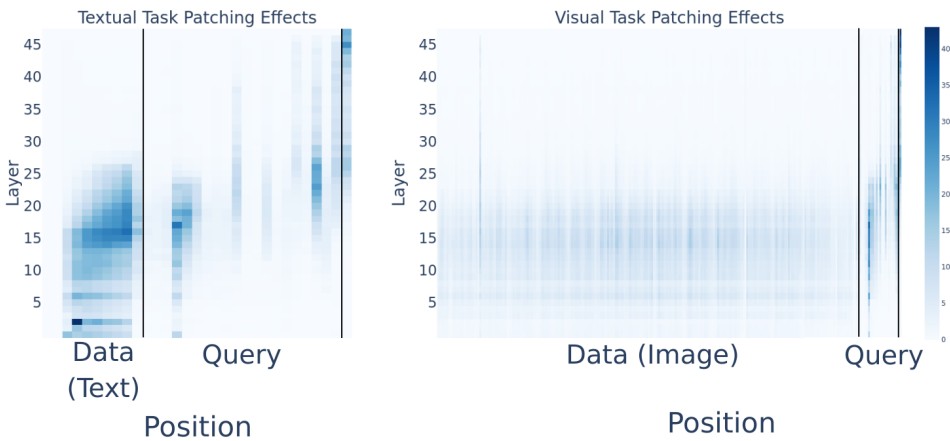

(c) **Patching effects for Gemma-3-12B for the textual (left) and visual (right) counting task.**

Figure 10: Patching effects for the counting task. The vertical lines split between data, query and generation positions.

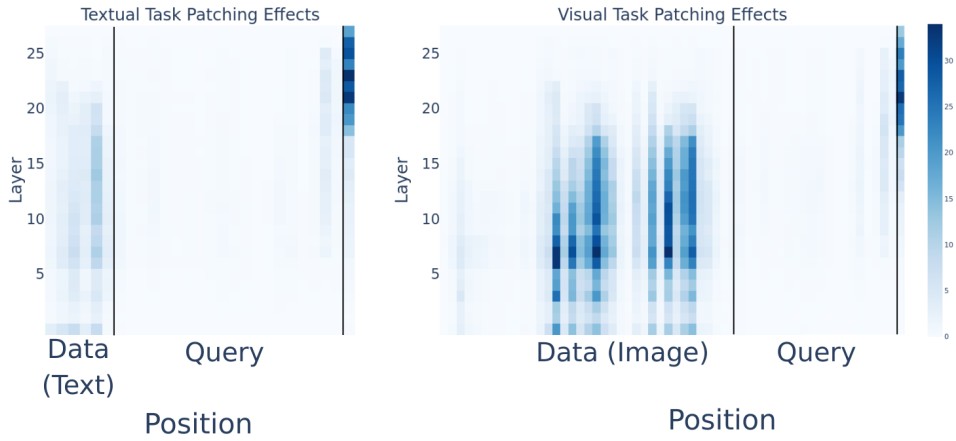

(a) **Patching effects for Qwen-7B-VL for the textual (left) and visual (right) arithmetic task.**

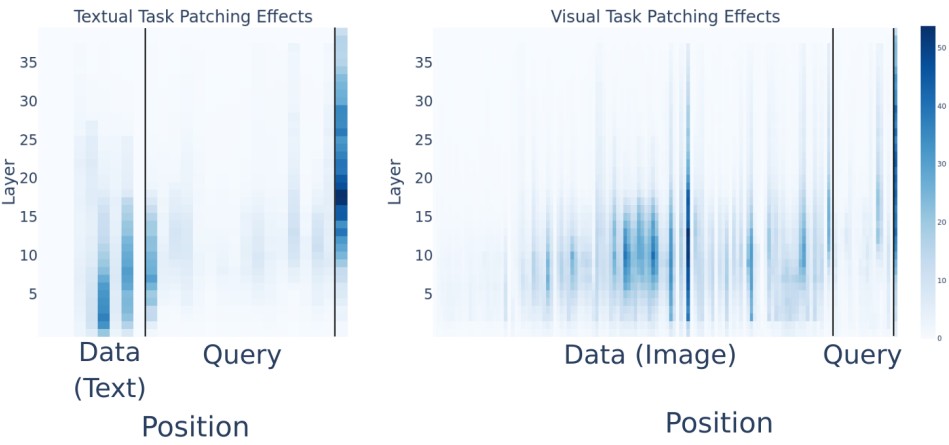

(b) **Patching effects for Pixtral-12B for the textual (left) and visual (right) arithmetic task.**

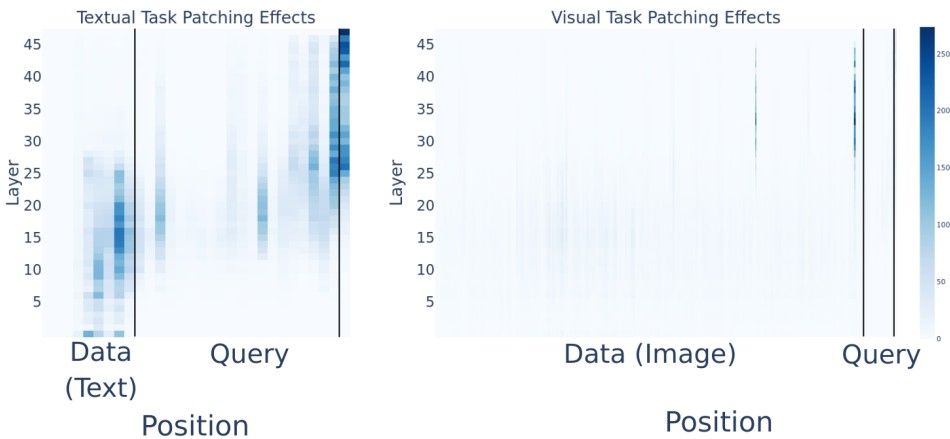

(c) **Patching effects for Gemma-3-12B for the textual (left) and visual (right) arithmetic task.**

Figure 11: Patching effects for the arithmetic task. The vertical lines split between data, query and generation positions.

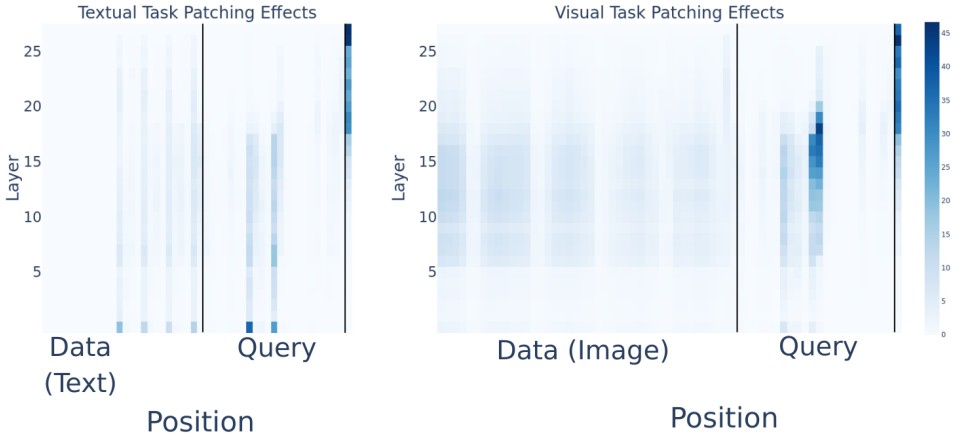

(a) **Patching effects for Qwen-7B-VL for the textual (left) and visual (right) spatial ordering task.**

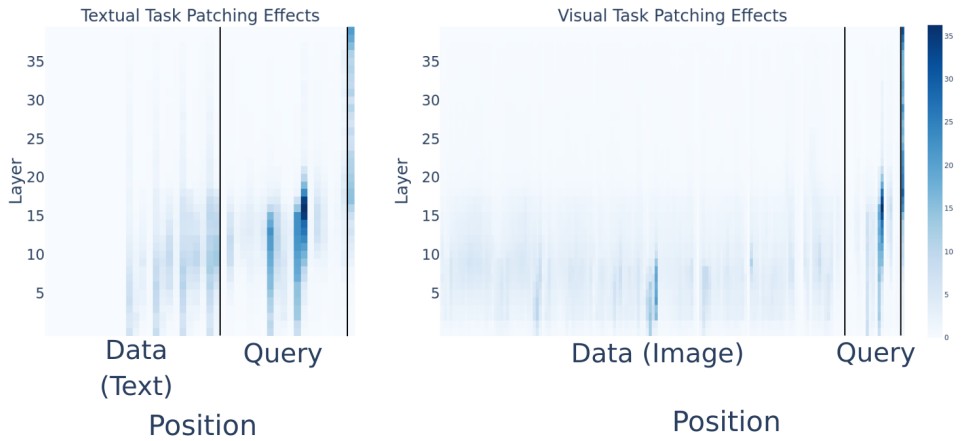

(b) **Patching effects for Pixtral-12B for the textual (left) and visual (right) spatial ordering task.**

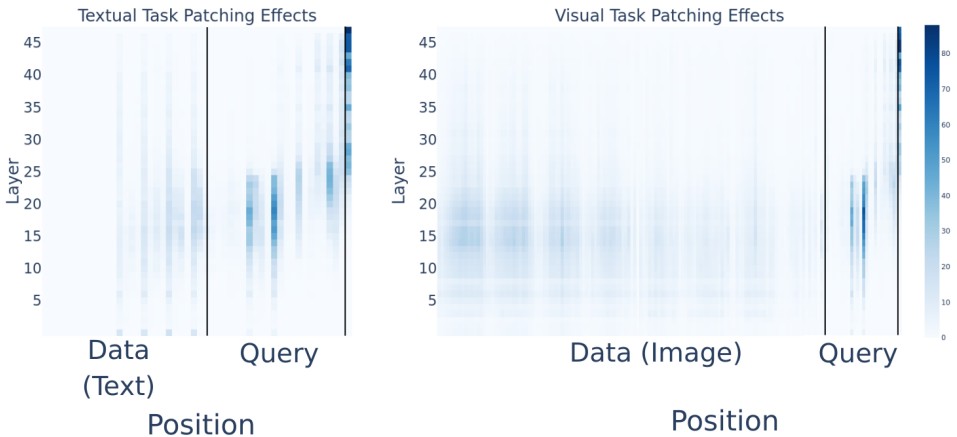

(c) **Patching effects for Gemma-3-12B for the textual (left) and visual (right) spatial ordering task.**

Figure 12: Patching effects for the spatial ordering task. The vertical lines split between data, query and generation positions.

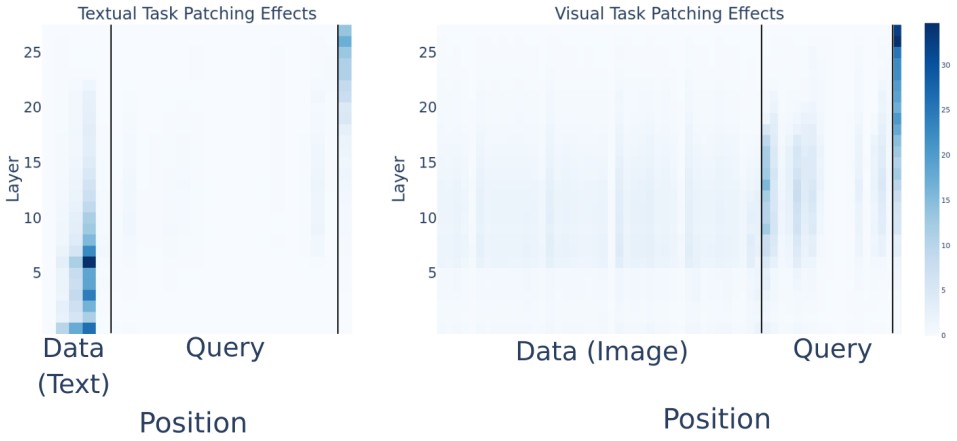

(a) **Patching effects for Qwen-7B-VL for the textual (left) and visual (right) factual recall task.**

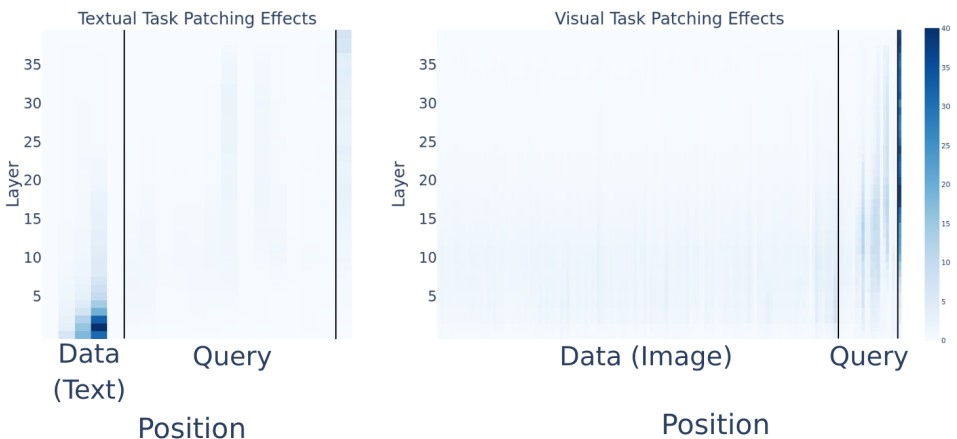

(b) **Patching effects for Pixtral-12B for the textual (left) and visual (right) factual recall task.**

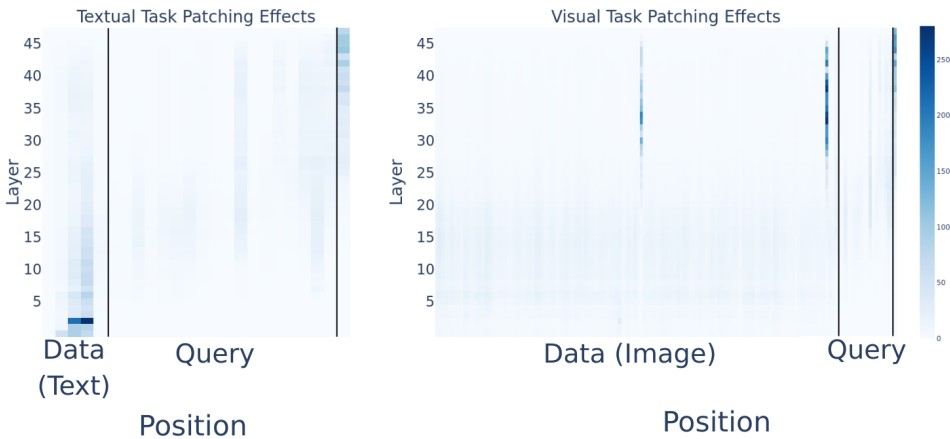

(c) **Patching effects for Gemma-3-12B for the textual (left) and visual (right) factual recall task.**

Figure 13: Patching effects for the factual recall task. The vertical lines split between data, query and generation positions.

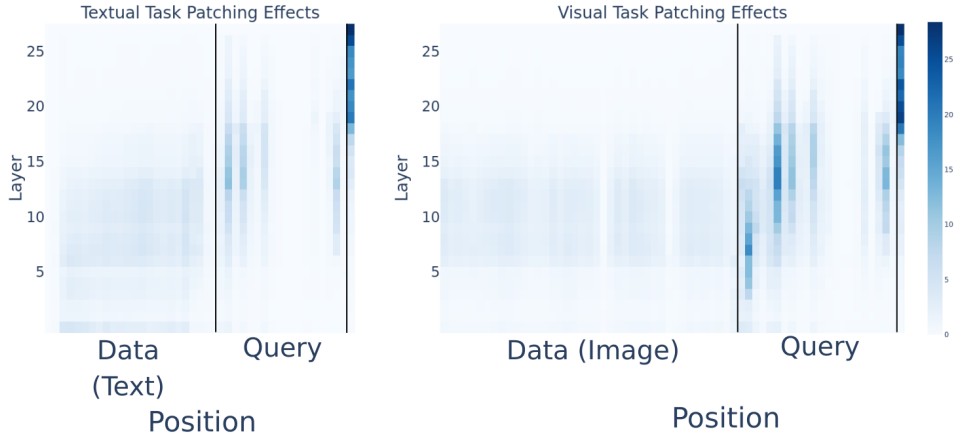

(a) **Patching effects for Qwen-7B-VL for the textual (left) and visual (right) sentiment analysis task.**

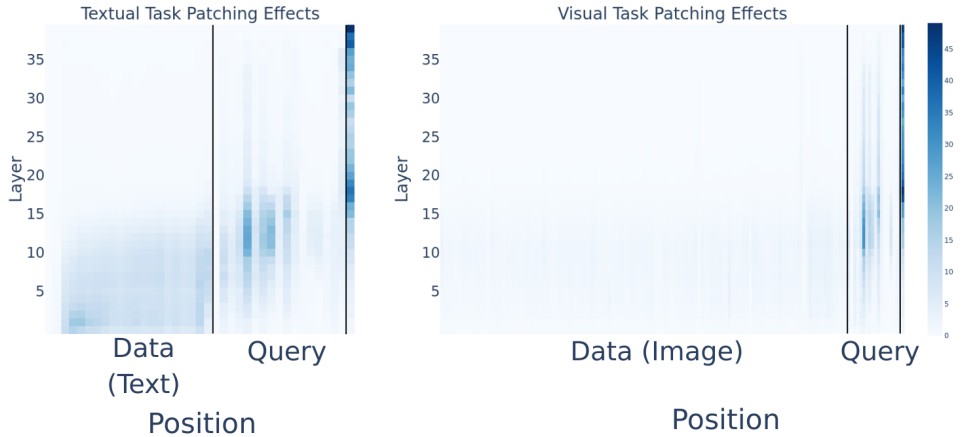

(b) **Patching effects for Pixtral-12B for the textual (left) and visual (right) sentiment analysis task.**

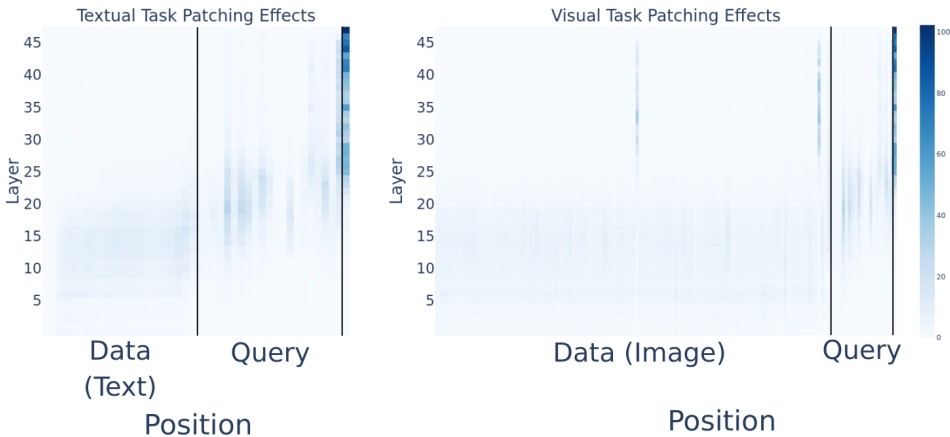

(c) **Patching effects for Gemma-3-12B for the textual (left) and visual (right) sentiment analysis task.**

Figure 14: Patching effects for the sentiment analysis task. The vertical lines split between data, query and generation positions.

