# OpenReview forum: "Same Task, Different Circuits: Disentangling Modality-Specific Mechanisms in VLMs"
_NeurIPS.cc/2025/Conference — NeurIPS 2025 poster_

### Official Review · Reviewer_KWdk · 2025-06-13

**Clarity:** 2
**Significance:** 3
**Originality:** 3
**Rating:** 5
**Confidence:** 3

**Summary:**

The paper introduces an improved method of identifying modality-specific processing circuits (attention heads and MLP output units that strongly influence the outputs) in VLMs. The paper treats the computations in VLMs as three stages: data, query, and generation. They found that while the circuits for data processing vary across modalities, the circuits for query and generation are largely the same for a given task. (Here, the circuits are a kind of averaged effect across the whole task dataset.)
The paper proposed to patch the high-level tokens back to lower layers when the VLM processes visual inputs. They showed that this method closes 32% of the gap between analogous visual and textual tasks.

**Questions:**

1. the authors might consider better explaining the relation between the observations and the proposed back-patching method. The authors might do the back-patching to both visual and textual inputs and contrast the improvement in both cases.
2. the authors might consider improving the clarity and the conciseness of the presentation.
3. I wonder if any of the non-adaptor architectures still have this property.

**Ethical Concerns:**

["NO or VERY MINOR ethics concerns only"]

**Final Justification:**

The paper offers a novel analysis that uncovers the drastic differences in the “circuits” of a VLM across different input modalities. Leveraging this, the authors propose a simple yet effective strategy—back-patching the high-level vision token into the input stream—that efficiently boosts performance and provides valuable insights for the VLM community.

My previous major concern was that the proposed strategy seems to be less relevant to the analysis. But with Appendix E.3 and the author's explanation, I am more confident in their relation.

My other concern was the clarity of some parts of the paper, which the authors have clarified and promised to improve.

The concerns of 1QgH are also addressed, which show the robustness of the observed performance improvement.

I personally think the concerns of KJ8j are less relevant in that the "real-world" datasets will have complex information that diminishes the power of controlled experiments, which is the core of this paper. Nevertheless, the performance improvement is also reassuring, though quite small.

**Limitations:**

yes

**Paper Formatting Concerns:**

I do have problems with selecting texts and clicking hyperlinks in the pdf file. I believe there is a problem with the formatting.

**Quality:**

3

**Strengths And Weaknesses:**

Strengths:
1. the perspective and analysis are novel, especially when applied to VLMs.
2. the circuit identification and comparison methods have some extent of novelty beyond the previous works. The components in the proposed methods have been carefully considered and controlled.
3. the swapping experiment is strong evidence for the main claim. This evidence confirms the validity of the proposed normalised IoU methodology and the results.
4. the proposed back-patching method is simple and effective.


Weaknesses:
1. I don't really immediately see why the proposed back-patching method is the logical solution to the circuit difference problem. For me, it is not surprising to see that the data circuits are different (but I am surprised that the query and generation circuits are not changing according to the input modalities), and therefore I don't understand the necessity to forcefully align the visual embeddings in the earlier layers to be more text-orientated. I wonder if the same technique can be used for textual inputs and we can still see performance improvement.
2. the presentation is good but not ideal. For me, there are some key steps with limited explanation. E.g., around line 170 the authors mentioned that the special visual tokens lead to the non-trivial positional mapping, but from my understanding, in the previous paragraph the authors already mentioned that they will not consider the data positions. Another example is that it seems the "patching effects" in figure 3 are never clearly defined (I have to assume that it refers to the "attribution patching" in section 2.2). A final example is that in line 229 the authors referred to another work without clearly explaining "unembedding matrix".

---

> ### Author Rebuttal · Authors · 2025-07-29
>
> Thank you for your detailed comments and feedback!
>
> We are glad that the novelty of our analysis and methodology came through clearly, and we appreciate the recognition of the strengths in our circuit comparison approach, the normalized IoU analysis, and the effectiveness of back-patching. Below, we address your concerns point by point.
>
> **1. "I wonder if the same technique (back-patching) can be used for textual inputs and we can still see performance improvement"**
>
> This is a great suggestion, which we indeed executed and reported in Appendix E.3 (referencing it in line 255). Since textual tokens are already “aligned with themselves”, gains here would reflect added compute. We find that visual back-patching yields larger improvements than textual back-patching in most settings, supporting our hypothesis that gains stem from improved alignment rather than additional depth alone. We will highlight this more clearly.
>
> **2. "I don't really immediately see why the proposed back-patching method is the logical solution to the circuit difference problem"**
>
> We appreciate your question, and want to clarify – the difference between circuits employed in the model to solve visual / textual tasks isn't necessarily a problem – it’s just a finding. This, in addition with our additional findings:
>
> (1) the difference lies mainly in the data positions; and
>
> (2) visual embeddings only converge towards the higher-performing textually-aligned embedding in later layers;
>
> leads us to utilize the aligned embeddings from later layers earlier in the model, to allow for higher performance with a simple intervention.
> Per your comment, we will better clarify this motivation in the introduction section.
>
> **3. "I wonder if any of the non-adaptor architectures still have this property."**
>
> Most open-weight SOTA VLMs use adapter-based architectures, including those we study. Whether our findings extend to non-adapter architectures is an important direction for future work, and we’ll highlight this in the limitations section.
>
> **4.** Regarding comments on presentation:
>
> * **"patching effects are not defined, I have to assume that it refers to the "attribution patching" in section 2.2"**: This is correct! We will point to section 2.2 in this figure to make this definition more clear.
>
> * **"in line 229 the authors referred to another work without clearly explaining "unembedding matrix"**: The unembedding matrix is another name for the language modeling head (or vocabulary projection), the last linear layer in the model that projects the embedding to the vocabulary space. Thank you for pointing this out - we will rephrase this line to avoid confusion.
>
> * **"the authors mentioned that the special visual tokens lead to the non-trivial positional mapping, but from my understanding, in the previous paragraph the authors already mentioned that they will not consider the data positions"**:  This non-trivial mapping happens in few cases where a single token is mapped to multiple tokens. One such example is a visual prompt that has a "<vision_end>" token and a "\n" following token and the analog textual prompt containing a single '"""\n' token which we map to both visual tokens. In this case, we consider these tokens to be query tokens (as they don't contain any visual information), and thus the mapping between query tokens isn't always one-to-one. We will make this part more clear in the revision.
>
> We hope our answers are satisfactory. Please let us know if you have any other questions or concerns.

---

> > ### Comment · Reviewer_KWdk · 2025-08-02
> >
> > Thanks for the response. All my concerns have been addressed.

---

### Official Review · Reviewer_1QgH · 2025-06-16

**Clarity:** 3
**Significance:** 2
**Originality:** 4
**Rating:** 5
**Confidence:** 3

**Summary:**

This paper investigates the accuracy gap for VLMs between the image-to-text inputs on tasks like counting objects/words or arithmetics by means of circuits, which are task-specific computational sub-graphs. They show that these circuits are disjoint between modalities primarily for data positions– especially so for Qwen2 & Pixtral – by means of the “Faithfulness” metric that uses counterfactual prompts and that in later layers the alignment between representations between text and visual tasks increases. Based on this finding, they propose Back-patching of visual tokens into earlier layers of the VLM, which can be applied post-hoc and show that this simple step improves performance on Counting, Arithmetic, Color Ordering, Factual Recall and Sentiment Analysis tasks for Qwen2, Pixtral, and Gemma2.

**Questions:**

For the most crucial points I would like to see addressed in this paper before giving a more beneficial rating, I refer to Weaknesses.

- How does the back-patching of visual tokens perform for VLMs with very strong multimodal capabilities like Pali-Gemma [1,2]?
- Would the authors expect the back-patching to lead to a higher focus of VLMs on image modality than on text modality as reported in, e.g., [3,4,5]?
    - An experiment showing this explicitly would greatly increase the value of the paper, especially if the compute tradeoffs are not beneficial for the back-patching.

**Citations**
- [1] PaliGemma: A versatile 3B VLM for transfer
src: https://arxiv.org/pdf/2407.07726
- [2] PaliGemma 2: A Family of Versatile VLMs for Transfer
src: https://arxiv.org/pdf/2412.03555
- [3] DO VISION & LANGUAGE DECODERS USE IMAGES AND TEXT EQUALLY?
HOW SELF-CONSISTENT ARE THEIR EXPLANATIONS?
src: https://arxiv.org/pdf/2404.18624
- [4] Visual cognition in multimodal large language models
src: https://www.nature.com/articles/s42256-024-00963-y
- [5] WHY CONTEXT MATTERS IN VQA & REASONING: SEMANTIC INTERVENTIONS FOR VLM INPUT MODALITIES
src: https://arxiv.org/pdf/2410.01690

**Ethical Concerns:**

["NO or VERY MINOR ethics concerns only"]

**Final Justification:**

The rebuttal convincingly addresses my main concerns. The authors demonstrate via bootstrapping that the improvements from back-patching are statistically significant in nearly all cases, strengthening the robustness of their claims. They also extend evaluation to the PaliGemma2 family across multiple scales, showing consistent gains and providing a plausible explanation for the unexpected 28B results. Finally, they clearly position back-patching as a proof-of-concept guided by mechanistic insights, rather than a ready-for-application method. With these clarifications, I believe the paper makes a meaningful and well-supported contribution, and I raise my score.

**Limitations:**

With the exception of the Limitations I stated in the Weaknesses section, the limitations are discussed.

**Quality:**

3

**Strengths And Weaknesses:**

## Strengths
The idea to improve the performance of VLMs by means of patching vision tokens into earlier layers of the VLM is, to my knowledge, novel and represents an elegant solution to improve the performance of VLMs for vision tasks by allowing the adapter to make better use of the powerful LLM it connects to.
- They also show that this does improve performance, even for models like Gemma-3, which showcases a much stronger cosine similarity as the multimodal aspect is much more enforced for these models than for the other 2 VLMs (Qwen2 and Pixtral).

The paper is clearly written, well-formatted, and presents its idea well, as indicated by:
- Overview Figures 1 & 2 give immediate intuition about the main concept of the paper.
- All figures have clear, descriptive descriptions with highlighted takeaways.
- Use of italics to highlight key messages

Rich in information:
The appendix in Supplementary Material contains a lot of relevant information and is well organized.
- Code in Supplementary Material is present and exhaustive (note: please make the README a little bit more descriptive)


## Weaknesses
**Evaluation does not incorporate scaling of models and compute cost** Evaluation is performed on 3 adapter models with one fixed size. One of their main findings is that image tokens from the adapter are assimilated by the LLM too late, so the natural question arises of whether this issue still persists for larger VLMs. It is crucial for potential practitioners to understand where back-patching is useful. Further, their patching approach increases the computational cost of inference, so this needs to be weighed against just using a larger model.

If it turns out that larger models are more efficient by having a better performance vs. compute-tradeoff than the proposed method, this significantly changes the contribution from a solution to a very interesting finding that requires more research before application.

To address this concern: Perform an evaluation where the same model family with different sizes is compared against back-patching, comparing the performance improvements and evaluating the inference cost.

**No Statistical Significance**  The authors correctly state in the checklist that neither error bars nor statistical significance are used.
 However, their justification would be valid for almost any machine learning paper and, therefore, more of an excuse: “All of our experiments are performed across all available data in a single execution. Thus, we have no standard deviation results to report.”

Especially since it is possible to obtain error bars by means of bootstrapping in case one does not wish to rerun the experiments and/or better use multiple seeds during evaluation for the generation over the entire dataset.

To address this concern: Show that the performance improvements using back-patching are statistically significant (e.g. by one of the previously mentioned methods).

**Nitpicks:**
- Please add a reference to the absolute Task Accuracies in Appendix C to the table description in Table 1.
- There appears to be something off with the format of the PDF (e.g., citations not clickable, text not highlightable); please fix this for the final version of the paper.

---

> ### Author Rebuttal · Authors · 2025-07-29
>
> Thank you for your valuable comments and feedback!
>
> We appreciate you recognizing the novelty, elegance and generality of our interpretability-guided improvement to VLM performance, and for finding our paper clear and well-presented. We address your comments one by one:
>
> **1. "No Statistical Significance results…To address this concern: Show that the performance improvements using back-patching are statistically significant"**
>
> Thank you for bringing this up. Per your suggestion, we show that back-patching leads to statistically significant performance improvement by running a bootstrapping experiment (1000 iterations, full sample size). The results, across all but one combination of model+task, show high significance confidence. The full bootstrapping results are in the following table (showing the clean accuracies, taken from Appendix C, and the bootstrapped mean and std of the post back-patching accuracy):
>
> | Model | Counting | Arithmetic | Spatial Ordering | Factual Recall | Sentiment Analysis |
> |---|---|---|---|---|---|
> | Qwen Clean | 70.9% | 67.6% | 74.2% | 68.1% | 92.6% |
> | Qwen Back-patched | 75.0% ± 1.9% | 76.9% ± 1.3% | 74.8% ± 0.6% | 69.7% ± 2.2% | 94.8% ± 1.4% |
> | Pixtral Clean | 66.1% | 22.7% | 80.6% | 45.0% | 69.7% |
> | Pixtral Back-patched | 68.3% ± 2.2% | 29.4% ± 1.6% | 84.7% ± 1.4% | 49.1% ± 2.3% | 87.4% ± 2.2% |
> | Gemma3 Clean | 73.4%  | 87.5%| 46.0% | 66.5% | 92.6% |
> | Gemma3 Back-patched | 75.4% ± 2.1% | 94.2% ± 0.7% | 47.5% ± 0.9% | 68.6% ± 1.8% | 98.0% ± 0.9% |
>
> To measure significance, we use a high confidence threshold of 0.99, and validate that the lower bound of the mean (which is the mean across 1000 iterations minus the error margin) is higher than the clean baseline accuracy. This shows all performance improvements are significant except for the Qwen+factural recall case, where back-patching leads to only a minor improvement.
> We will add these results to Table 1 (back-patching results), describe the bootstrapping procedure in the revision's appendix, and add the relevant code.
>
> **2. "Evaluation does not incorporate scaling of models…To address this concern: Perform an evaluation where the same model family with different sizes is compared against back-patching, comparing the performance improvements"**
>
> To incorporate evaluation across scales, as well as answer your later question on the back-patching effect on PaliGemma, we perform the back-patching experiment on PaliGemma2 models, at 3 model sizes - 3B, 10B and 28B.
> The results, presented in the following table, show that across model sizes, back-patching is a valid fix to the "lack of visual alignment" problem, showing relative performance improvements.
>
> | Model | Counting | Arithmetic | Spatial Ordering | Factual Recall | Sentiment Analysis |
> |---|---|---|---|---|---|
> | PaliGemma2-3B-224 Clean | 64.0% | 0.7% | 15.4% | 32.7% | 48.7% |
> | PaliGemma2-3B-224 Back-patched | 65.3% ± 2.9% |  32.3% ± 3.8% | 23.5% ± 3.4% | 39.2% ± 3.9% | 56.3% ± 3.9% |
> | PaliGemma2-10B-224 Clean | 66.0% | 8.7% | 14.0% | 38.7% | 56.7% |
> | PaliGemma2-10B-224 Back-patched | 69.4% ± 3.1% | 36.8% ± 4.1% |  25.9% ± 3.4% | 56.2% ± 4.0% | 72.8% ± 3.6% |
> | PaliGemma2-28B-224 Clean | 22.0% | 6.7% | 12.0% | 30.7% | 10.7% |
> | PaliGemma2-28B-224 Back-patched | 26.2% ± 3.6% | 30.8% ± 3.8% | 15.5% ± 2.9% | 36.0% ± 4.0% | 26.0% ± 3.6% |
>
> Across most tasks and scales, back-patching improves performance by a significant margin, across scales. The bootstrapping settings are identical to the ones in the previous table.
>
> While back-patching isn't a perfect solution across settings, *we don't view it as such*. To quote the reviewer, we view it as "a very interesting finding that requires more research before application" -- it's a PoC of a potential way to bridge a limitation of VLMs, based on the mechanistic insights which are the main focus of our paper.
>
> Nevertheless, we will add this evaluation of back-patching across scales and its implications to the revised paper.
>
> **3. “their patching approach increases the computational cost of inference, so this needs to be weighed against just using a larger model.”**
>
> We treat back-patching as a PoC based on mechanistic insights (which are the main focus of our paper) rather than a ready solution. With that said, the additional experiments on the PaliGemma family show that this method can be applied on models of different scales to achieve performance improvements. Thus, we argue that such interpretability-based insights can provide improvements orthogonally to using larger models, which can be limited by availability.
>
>
> **4. "Would the authors expect the back-patching to lead to a higher focus of VLMs on image modality than on text modality as reported in, e.g., [3,4,5]?"**
>
> This is an interesting question. We speculate back-patching doesn't affect the amount of attention given to image positions, but mainly allows the propagation of more "concise" and textually-aligned information from image tokens from early layers. However, further analysis into the effects and limitations of back-patching on visual token processing is an interesting direction. As we view the main focus of our work in the mechanistic analysis, and view back-patching as a finding rather than a perfect solution, we leave these investigations to future work.
>
> **5.** We will fix the two additional comments (reference to Appendix C from Table 1, PDF formatting issues) in the revision of the paper.
>
> To summarize, we wish to re-iterate that we don’t view back-patching as a limitation-free solution to the performance gap, but rather a fix showing that interpretability-guided insights can help bridge modality performance gaps.
>
>
> We hope our updated results and answers are satisfactory. Please let us know if you have any other questions or concerns.

---

> > ### Comment · Reviewer_1QgH · 2025-08-04
> >
> > Thanks for the detailed response.
> > My main concerns have been addressed.
> >
> > I have only two more points:
> >
> > ---
> >
> > I would much appreciate it if the argument for **3.** would be clearly stated in the Limitations.
> > - That you treat back-patching as a Proof-of-Concept rather than a ready solution.
> >
> >
> > ---
> >
> > Further, I have one Question regarding the results table for  **2.**
> > - Do you have any hypotheses why the 28B model showcases a general drop in performance compared to the smaller model?
> >     - Especially for counting and sentiment analysis.
> >     - In table 14 of the official [PaliGemma2](https://arxiv.org/pdf/2412.03555) the trend seems to clearly indicate a general performance improvement for larger models.

---

> > > ### Author Response · Authors · 2025-08-04
> > >
> > > Thank you for engaging in the rebuttal. To answer the two points you brought up:
> > >
> > > 1. Per your request, we will emphasize in our limitations section that back-patching is a PoC based on our main mechanistic findings, rather than a ready-for-application solution.
> > >
> > > 2. This is a great question that also made us wonder and double-check our results. We looked at the PaliGemma2 paper (specifically at the figures showing results for the 28B size - Fig4, Fig5, Table 13, Table 14). We noticed that in Figure 4, which shows results across a wide range of benchmarks, the scaling of PaliGemma to 28B sometimes only leads to minor improvements, and in some cases even leads to performance degradation, as also stated by the authors of that paper at the end of Page 4 (in section 4.1.1). This is also visible under some settings in Fig5, Table 13 and Table 14.  The authors of PaliGemma2 hypothesize that it might be due to "the underlying Gemma 2 27B model is trained from scratch, as opposed to the 2B and 9B models, which are distilled" (end of section 4.1.1).
> > > We thus hypothesize that our counting & sentiment tasks might also suffer from this lack of transferability shown in PaliGemma2-28B. We will present this discussion in the added appendix on back-patching across scales.

---

> > > > ### Comment · Reviewer_1QgH · 2025-08-04
> > > >
> > > > I thank the authors for their answer. All my concerns have been addressed. I will also raise my score accordingly.
> > > >
> > > > I would encourage the authors also to add this explanation for the results of PaliGemma2-28B next to the presented results in the final version of the manuscript.

---

### Official Review · Reviewer_KJ8j · 2025-07-03

**Clarity:** 2
**Significance:** 2
**Originality:** 3
**Rating:** 3
**Confidence:** 3

**Summary:**

The authors investigate why vision–language models (VLMs) are consistently less accurate on visual than on textual versions of the same task. Based on prior works on circuit analysis (minimal subset of connected model components required for a specific task), the authors extract the circuits across three VLMs. The analysis shows that vision and language circuits are largely structurally disjoint—yet the sub‑circuits handling the query and answer generation are functionally interchangeable; the accuracy gap stems mainly from modality‑specific data‑processing components. In addition, visual token representations align with their textual analogues in late layers. Based on these observations, the authors propose an inference-time method that "back-patches" these late-layer visual activations into earlier layers, and demonstrate its effectiveness experimentally.

**Questions:**

**Rationale for circuit‑level analysis**
  - The study follows the circuit‑extraction procedure of Mueller et al. (2025), isolating a minimal, interconnected sub‑graph that is causally sufficient for each task. Could you clarify why this circuit‑based lens is preferable to more common attribution techniques—e.g., gradient‑ or perturbation‑based importance scores—when diagnosing modality gaps? What limitations of those alternatives (faithfulness, resolution, inter‑head interactions, etc.) motivated your choice? Or do you view circuit extraction as just one of several equally valid approaches?

**Evaluation beyond toy settings**

 - Counting, spatial ordering and factual recall are established tasks, but the visual sentiment analysis and two‑operand arithmetic are not realistic. Overly templated captions rarely occur in practice. Likewise, restricting prompts to single‑word answers and short captions departs from real VQA workloads.

 - Can authors evaluate the effectiveness of back-patching on a couple of common VQA benchmarks that have paired realistic image-texts (e.g., object counting can also be done based on images with dense captions such as [1])?

[1] Urbanek et al., A Picture is Worth More Than 77 Text Tokens: Evaluating CLIP-Style Models on Dense Captions, CVPR 2024

**Ethical Concerns:**

["NO or VERY MINOR ethics concerns only"]

**Final Justification:**

After discussing with the authors and reviewing other reviewers' opinions, I believe the writing of the original paper needs significant improvement for clarity. Multiple sections could be made more accessible. I would be disappointed if the manuscript were to remain largely unchanged. However, it would be great if the authors make the modifications they promised in the rebuttal.

My major concern is that the applicability of the method is very restricted to cases where the answer is a single token, and prompts need to be arranged in a very specific way for analysis (L117–123). All experiments in the paper are conducted on relatively simple, toy-like visual datasets (e.g., simple object counting, sentiment analysis). This also explains why the baseline performance on more **realistic tasks**, such as RealWorldQA and VQAv2, is much lower than publicly reported results.

For this reason, I take the claimed effectiveness of the back-patching method with a grain of salt. In their rebuttal, the authors claim this limitation is a fundamental issue in circuit analysis and not something they aim to solve in this work. Nevertheless, more experiments on real-world datasets are necessary to properly justify the empirical contribution of this paper.

That said, I agree with other reviewers that the analysis itself is interesting. I hope this justifies my rating: "3: Borderline reject – Technically solid paper where reasons to reject, e.g., limited evaluation, outweigh reasons to accept, e.g., good analysis."

**Limitations:**

yes

**Quality:**

2

**Strengths And Weaknesses:**

Strength
- The empirical findings of the paper are interesting and show new insights.
- The ablation studies and analysis part are carefully designed.

Weakness
- I am concerned about the validity of the findings beyond simple toy datasets. All experiments are run on highly simplified, synthetic datasets. Tasks such as two‑operand arithmetic or sentiment judgments over short, templated captions have little real‑world relevance, and the captions themselves are far simpler than those found in practical VQA settings. As a result, it is unclear whether the conclusions—and the proposed late‑layer “back‑patching” fix—would hold on more complex images, free‑form captions, or open‑ended answers.

- Prompting setup is also restricted. Circuit discovery is performed only on prompts whose ground‑truth answer is a single word, further distancing the evaluation from realistic multimodal QA scenarios.

- The writing of the paper needs to be improved for better clarity and accessibility. In particular, the Preliminaries Section could be made significantly more accessible. For example,

  - Formula (1) appears without motivation (e.g., why k=5?) or derivation, and obscures rather than illuminates the Hanna et al. (2024) method.

  - Key terms such as “textual task,” “visual task,” and how prompts are split into “discovery” and “evaluation” subsets are introduced without definition or concrete examples. It would be better to expand this paragraph.

I would suggest authors to polish these sections, add illustrative examples, and remove unnecessary formalism.

---

> ### Author Rebuttal · Authors · 2025-07-30
>
> Thank you for the feedback!
>
> We appreciate you finding our findings interesting and insightful, as well as mentioning the thoroughness of our ablations.
> We address your comments and questions one by one:
>
> **1.** Regarding your comments and questions on evaluation beyond toy settings:
>
> **"Can authors evaluate the effectiveness of back-patching on a couple of common VQA benchmarks that have paired realistic image-texts (e.g., object counting can also be done based on images with dense captions such as [1])?"**
>
> Thank you for this suggestion. We have performed an evaluation of back-patching on general vision-question answering using the VQAv2 dataset [9] and the more up-to-date RealWorldQA [10] dataset, two common visual-question answering benchmarks. In the following table, we report the results, showing that *back-patching textually-aligned visual embeddings is effective even in general VQA prompts*:
>
> | | Qwen2-7B-VL | Pixtral-12B | Gemma3-12B |
> |---|---|---|---|
> | VQAv2 Clean | 65.0% | 62.2% | 65.7% |
> | VQAv2 Back-patched | 68.2% ± 0.9% | 66.2% ± 1.1% | 70.3% ± 1.3% |
> | RealWorldQA Clean | 58.8% | 58.3% | 59.1% |
> | RealWorldQA Back-patched | 61.6% ± 1.0% | 60.1 ± 1.2% | 60.0% ± 0.8% |
>
> The RealWorldQA results are calculated on the entire dataset. The VQAv2 results are calculated as the average of 3000 randomly sampled visual prompts (without image repetition). We limit the analysis to 3000 prompts due to computational requirements of exploring different back-patching settings. Bootstrapping was performed (similarly to the settings in the response to reviewer 1QgH) for 1000 iterations with full sample size to validate statistical significance.
>
> We will add these results to the appendix and update the code in the revision.
>
> **"Restricting prompts to single‑word answers and short captions departs from real VQA workloads"**,
> **"Circuit discovery is performed only on prompts whose ground‑truth answer is a single word"**
>
> While this limits analysis approach, this stems from an inherent limitation of circuit discovery methods (as we mention in lines 121-122). Such methods (as in [1,2,3,4,5], and more) are all limited to identifying model components crucial to the completion of a single token. Extending these methods to analyze the completion of a complex, multi-token completion isn't trivial and *is an active research question in mechanistic interpretability*. Additionally, our circuit intersection analysis relies on the positional alignment of query tokens, making it less feasible to be applied on prompts with different-length questions.
>
> While this limits the capabilities of *all* circuit discovery research, we follow standard procedures ([3,6,7,8] and more) to isolate the behavior we are localizing, such that it could be expressed in a single token containing the answer, and will allow us to identify the key components for each task, which we further analyze.
>
> We will add a discussion on this point to a revised limitations section.
>
> **2. Rationale for circuit‑level analysis: "Could you clarify why this circuit‑based lens is preferable to more common attribution techniques—e.g., gradient‑ or perturbation‑based importance scores—when diagnosing modality gaps?"**
>
> This is an interesting question. Could you provide specific references or methods? As “gradient- or perturbation-based importance scores” could refer to different approaches.
>
> If you mean input attribution techniques (identifying influential input features), these are valid and provide a different perspective for investigating modality performance gaps. We chose circuit extraction because it reveals the computational pathways that process different modalities, allowing us to characterize the role of different model components and how they interact.
> This structural lens enabled our key analyses: examining circuit intersections (Section 4.1) and testing component interchangeability between modalities (Section 4.2) to identify similar or different functionalities.
> By isolating causally sufficient subgraphs, we can understand how different modalities utilize shared versus distinct computational mechanisms.
>
> If you mean component attribution techniques (scoring individual model components), we should clarify that our approach actually incorporates these methods. We use attribution patching with integrated gradients [1,4], which is activation-based and gradient-based, to score component importance and build sufficient circuits from high-importance components.
>
> We view our circuit-based analysis as one rigorous lens among several valid methodological approaches. If you have any further questions in this manner, we would love to elaborate.
>
>
> **3.** Writing comments:
>
> **"Formula (1) appears without motivation (e.g., why k=5?) or derivation, and obscures rather than illuminates the Hanna et al. (2024) method."**
>
> Thank you for pointing this out. We wish to clarify: Formula (1), showing the importance metric we measure per model component, was taken almost as-is from Hanna et al. [4], and is a standard circuit discovery metric. The number of integrated gradient steps (k) is chosen to be 5 in the original work [4] to be a good tradeoff between accuracy and runtime.
> Per your comment, we will add a relevant section in the appendix to expand on this method instead of merely citing it.
>
> **"Key terms such as “textual task,"visual task,” and how prompts are split into “discovery” and “evaluation” subsets are introduced without definition or concrete examples. It would be better to expand this paragraph."**
>
> Regarding textual vs visual tasks - we present a quick example of textual and visual tasks in the abstract (e.g., an analog pair of a visual and textual tasks is "counting a specific object in an image" versus "counting a specific word in a word sequence"), but we realize this should be re-iterated. We will add such an example to the intro and connect to it in the caption of Figure 2.
>
> Regarding discovery vs evaluation subsets - we explain the division to discovery and evaluation datasets in Appendix B.3. We will add a reference to it to Section 2.2., where these split is first presented.
>
> Per your request, we will polish the preliminaries section to improve its presentation and remove unnecessary formalism.
>
> .
>
> We hope our updated results, answers and fixes are satisfactory. Please let us know if you have any other questions or concerns.
>
>  References:
>
> [1] N Nanda, "Attribution Patching: Activation Patching At Industrial Scale", 2022.
>
> [2] F Zhang, et al., "Towards Best Practices of Activation Patching in Language Models: Metrics and Methods", 2023
>
> [3] K Wang, et al., "Interpretability in the wild: a circuit for indirect object identification in gpt-2 small", 2022
>
> [4] M Hanna, et al., "Have Faith in Faithfulness: Going Beyond Circuit Overlap When Finding Model Mechanisms", 2024
>
> [5] E Ameisen, et al., "Circuit Tracing: Revealing Computational Graphs in Language Models", 2025
>
> [6] M Hanna, et al., "How does GPT-2 compute greater-than?: Interpreting mathematical abilities in a pre-trained language model", 2023.
>
> [7] N. Prakash et al., "Language models use lookbacks to track beliefs", 2025.
>
> [8] A. Stolfo et al., "A Mechanistic Interpretation of Arithmetic Reasoning in Language Models using Causal Mediation Analysis", 2023.
>
> [9] Y. Goyal et al., "Making the V in VQA Matter: Elevating the Role of Image Understanding in Visual Question Answering", 2017.
>
> [10] X.AI. Grok-1.5 vision preview. https://x.ai/blog/grok-1.5v, 2024a. 9, 15
>
> [11] A. Mueller et al., "MIB: A Mechanistic Interpretability Benchmark", 2025

---

> > ### Author Response · Authors · 2025-08-06
> >
> > We again thank the reviewer for their feedback. As we approach the end of the discussion period, we wanted to verify that our comment has answered all of the reviewer's questions. We welcome any remaining questions and would appreciate confirmation on whether our responses have sufficiently addressed your concerns.

---

> > ### Comment · Reviewer_KJ8j · 2025-08-08
> >
> > Thanks authors for the response. It really helped me gain a better understanding of the work. I highly encourage the authors to improve the clarity and readability of the relevant sections, as demonstrated in the response above. I have two follow-up questions:
> >
> > Q1: The performance shown on RealWorldQA appears to be much lower than publicly available results. Could the authors explain this discrepancy?
> >
> > In the response, it is stated that "The RealWorldQA results are calculated on the entire dataset." However, the reported performance [1] for Qwen2-7B-VL (68.5%) and Pixtral-12B (65.4%) seems much higher than what is shown in the response above.
> >
> > Q2: I’m curious—if you extend the tasks beyond simple arithmetic or object counting to more realistic ones like visual reasoning in diagrams (e.g., MathVista), is it still applicable/effective?
> >
> > [1] https://huggingface.co/spaces/opencompass/open_vlm_leaderboard

---

> > > ### Author Response · Authors · 2025-08-08
> > >
> > > Thank you for engaging in the discussion. To answer your follow-up questions:
> > >
> > > **1. The (baseline) performance shown on RealWorldQA appears to be much lower than publicly available results. Could the authors explain this discrepancy?**
> > >
> > > Thank you for mentioning this.
> > > Looking deeper into Qwen2, the performance difference likely stems from a difference in prompting: the publicly reported results for Qwen2-7B-VL allow extended responses with many tokens (as shown in qualitative examples in Appendix A of the Qwen2 paper [2]),  while in our setup (exemplified in our Appendix B.1) we force the model to answer immediately, which understandably leads to lower results.
> > > This setup was made due to compute constraints: as we mention in lines 238-248, we systematically search for the optimal back-patching layers per task, which is very compute intensive to perform for long-form completions.  While this limits direct benchmark comparison, it ensures consistency across our experimental framework.
> > >
> > > To address this comment:
> > > 1. We will clarify this setup difference in the new appendix we added on back-patching for general VQA.
> > > 2. Following your comment, we are experimenting with back-patching on *long-form* general VQA benchmarks using the best-performing back-patching setting for *single-token* completions. These results will give a sense on the usefulness of back-patching for long form content, without requiring extreme amounts of compute, and will be added to the general VQA appendix (probably won't be ready by the end of the discussion period, but will be added to the next revision). We wish to clarify (similarly to our response to reviewer 1QgH) that we don't view back-patching as a limitation-free solution to the performance gap across *all experimental settings*, but rather a fix showing that interpretability-guided insights can help bridge modality performance gaps.
> > >
> > >
> > > **2. if you extend the tasks beyond simple arithmetic or object counting to more realistic ones like visual reasoning in diagrams (e.g., MathVista), is it still applicable/effective?**
> > >
> > > This is an interesting question.
> > > Results from IsoBench [1], that contains similar tasks to MathVista (where the model answers questions based on a mathematical function visualization) also show that VLMs perform worse for visual inputs compared to textual inputs (where the input is phrased as an equation) - As seen in Section 2.1 in [1]. This performance gap makes us hypothesize that similar circuit similarities and differences will be seen in such tasks as well.
> > >
> > > However, given the lack of positional alignment between MathVista prompts and the before-mentioned current limitations of circuit analysis, we couldn't perform similar analysis on such prompts.
> > > There is very recent work [3] that aims to tackle these problems by allowing circuit analysis on single prompts, but it is still heavily limited (not applicable to VLMs, requires compute-heavy training to apply for a new model).
> > > It would be very interesting to re-visit this analysis once such methods allow for the analysis of mis-aligned positional circuits in VLMs as well.
> > >
> > >
> > > References:
> > >
> > > [1] D. Fu et al., "IsoBench: Benchmarking Multimodal Foundation Models on Isomorphic Representations", 2024.
> > >
> > > [2] P. Wang et al., "Qwen2-VL: Enhancing Vision-Language Model’s Perception of the World at Any Resolution", 2024.
> > >
> > > [3] E. Ameisen et al., "Circuit Tracing: Revealing Computational Graphs in Language Models", 2025.

---

> > > > ### Comment · Reviewer_KJ8j · 2025-08-09
> > > >
> > > > Thanks for the clarification. This addressed my questions

---

### Official Review · Reviewer_NpXT · 2025-07-06

**Clarity:** 3
**Significance:** 3
**Originality:** 4
**Rating:** 5
**Confidence:** 4

**Summary:**

This paper investigates the performance gap between vision and language tasks in VLMs by examining the underlying circuits for analogous problems (e.g., counting objects in an image vs. words in a text). The authors use causal analysis to identify task-specific circuits, revealing that vision and language modalities rely on structurally disjoint components. Despite this structural separation, they find that the sub-circuits responsible for processing the query and generating the answer are functionally interchangeable. The critical difference lies in the initial data integration, which is modality-specific. The authors hypothesize that the performance gap arises because visual representations align with their textual counterparts too late in the model's processing depth. As a remedy, they leverage back-patching -- a training-free intervention that injects these late-stage, text-aligned visual representations into earlier layers, to meaningfully reduce the performance gap.

**Questions:**

These are points I was curious about that could be interesting to discuss in the paper:

* The authors may want to check out recent work by Campbell (2024) which looks at VLM failures on similar tasks from the perspective of the "binding problem" in cognitive science, particularly regarding parallel vs. serial processing. In many ways, this paper's findings could be interpreted as showing that VLMs lack the deep, sequential processing pathways for vision that are naturally afforded by language. I am speculating here, but I would be interested to hear how the authors' work ties into that literature.
*  It would be very interesting to see the results of an experiment where the order of the query and the data in the prompt is swapped. This would likely change how information is integrated throughout the network and could impact the degree to which circuits are interchangeable and back-patching is effective.
* Figure 7 shows a different pattern of activation similarity for Gemma compared to the other models. Its visual-textual alignment is high from very early layers. Could this be due to Gemma being trained from scratch on multimodal data? This seems to contradict the general hypothesis that the model first aligns image data into a textual representation. Despite this, back-patching is still effective for Gemma. A discussion of this would be a welcome addition to the paper.

**Ethical Concerns:**

["NO or VERY MINOR ethics concerns only"]

**Final Justification:**

I thank the authors for addressing my concerns. I enjoyed reading the paper, and believe that the changes the authors plan to make will make the paper stronger. For these reasons, I will maintain my positive rating.

**Limitations:**

Yes.

**Quality:**

3

**Strengths And Weaknesses:**

**Strengths**

In general, I think this paper is very strong. The research question is well-motivated, the methodology for identifying and comparing circuits is compelling, and the proposed back-patching intervention is both simple and effective. The work builds upon a rich and relevant body of knowledge and doesn't try to reinvent the wheel. My review should be caveated with the fact that I am not an expert in mechanistic interpretability, so I have assumed that the application of methods like AP-IG and the faithfulness metric are consistent with standard conventions in the field.

**Weaknesses**

I would strongly shy away from framing the analysis around Marr's three levels. While I am a fan of Marr's framework, its introduction here (Line 140) seems to open a Pandora's box without adding significant value to the paper's core results. The paper doesn't elaborate on what Marr's levels are or how the structural and functional roles of circuits map onto that hierarchy. I assume the authors intend to map their analysis onto the implementational and algorithmic levels, respectively, but I don't believe it's that clean of a mapping. In fact, one could argue the entire analysis resides at the implementational level (i.e., identifying that different hardware implements the same algorithm/function). Removing the reference would improve the paper's focus and sidestep potential debates that detract from the empirical results.

The writing is not the most polished and seems rushed, which is a shame given the nice results. For instance:

* On line 72, the notation $LD(r,r')$ is used before the term logit distance is explicitly stated. Introducing the term first would improve readability.
* On line 83, the authors begin referring to circuits as $c$ without explicitly defining the notation (e.g., "a circuit $c$ is a set of components..."). This should be formalized.
* There is a quotation mark bug on line 163.
* The bar order in Figure 6 doesn't seem to match the legend, which can be confusing for the reader. While a quick check suggests they might be in the correct order, ensuring this is maximally intuitive is important. Furthermore, Figure 3, which illustrates the motivation for the sub-circuit analysis, is introduced very early and might be better placed closer to its detailed discussion in the text to improve the paper's flow.

---

> ### Author Rebuttal · Authors · 2025-07-29
>
> Thank you for your thoughtful and constructive feedback!
>
> We're glad to hear that you found our paper to be very strong, with the motivation, methodology and impact of our work coming through clearly.
>
> To address your comments and questions:
>
> **1. "I would strongly shy away from framing the analysis around Marr's three levels"**
>
> In mentioning Marr’s levels of analysis, our intent was to loosely connect structural and functional circuit analysis with the implementational and algorithmic levels, respectively (as you correctly assumed).
> However, we acknowledge that this mapping is ambiguous and not essential for understanding our findings. We will remove this reference entirely in the revision to maintain focus.
>
> **2. “The authors may want to check out recent work by Campbell (2024)”**
>
> We appreciate the reference to Campbell et al. 2024 [1], which we agree seems very relevant to our findings. Our work highlights that visual data representations align with their textual counterparts too late, which can be seen as a form of representational interference. Back-patching effectively injects aligned representations back into early layers of the model, functionally reducing this interference, similar in spirit to introducing more serial processing.
>
> **3. "It would be very interesting to see the results of an experiment where the order of the query and the data in the prompt is swapped."**
>
> We agree that reversing the order of the query and data might affect how information is integrated and may impact interchangeability and back-patching performance. However, some current VLMs are typically trained with the query following the image-text input, so reversing the order constitutes a distributional shift. While Qwen and Gemma 3 models do allow this reversal with some accuracy, we chose to focus on the more prevalent setting to match that of Kaduri et al., 2024 [2].  We will add this point to our limitations section.
>
> **4. "different pattern of activation similarity for Gemma…Could this be due to Gemma being trained from scratch on multimodal data?"**
>
> This is a great question, which we discuss in Appendix E.2. While Figure 7 shows that Gemma’s activation similarities increase in earlier layers than other models, the best back-patching layers (presented in Appendix E.2) match this pattern, showing that in most tasks, back-patching achieves the best performance when patching embeddings from higher-similarity layers back into lower-similarity layers (where in Gemma3, these layers indeed come earlier in the model, perhaps due to it being trained from scratch on multimodal data, as you suggest). We will add an additional reference to this discussion appendix around lines 233-234 to make it more clear.
>
> **5.** We appreciate the detailed suggestions for improving the clarity of the paper, and we will address these issues in the revision of the paper. Specifically - we will improve the definition of logit difference in line 72, fix quotation mark bugs, and make the order of bars in figure 6 more intuitive by showing a minimal bar when the faithfulness is near-zero so the order will be clear.
>
>
> We hope our answers are satisfactory. Please let us know if you have any other questions or concerns.
>
> References:
>
> [1] D. Campbell, Understanding the Limits of Vision Language Models Through the Lens of the Binding Problem, 2024
>
> [2] O. Kaduri et al., What's in the Image? A Deep-Dive into the Vision of Vision Language Models, 2024

---

> > ### Author Response · Authors · 2025-08-06
> >
> > We again thank the reviewer for their feedback. As we approach the end of the discussion period, we wanted to verify that our comment has answered all of the reviewer's questions. We welcome any remaining questions and would appreciate confirmation on whether our responses have sufficiently addressed your concerns.

---

### Decision · Program_Chairs · 2025-09-17

**Decision:**

Accept (poster)

**Comment:**

The submission focuses on investigating the performance gap between vision and language tasks in vision-language models, which is achieved by examining the underlying circuits for analogous problems in the corresponding data modality. The authors observed that while the sub-circuits are functionally interchangeable, the initial data integration is modality-specific. The authors then proposed "back-patching", which is a training-free intervention approach to inject text-aligned visual representations into earlier layers, which indeed reduce the modality performance gap.

The submission received three accept and one borderline reject ratings after rebuttal. Reviewers generally appreciate the method and analysis from the submission as well-motivated, novel, and insightful. Reviewer NpXT provided valuable feedback on the presentation of the submission, and found all of their concerns resolved after the rebuttal. Reviewer KWdk had questions about the connection between the analysis and the proposed remedy, which was also mostly addressed during rebuttal. There were also concerns on the empirical evaluations: Reviewer 1QgH requested model scaling experiments and statistical significance analysis, which were provided by the authors. 1QgH found them to be convincing, but requested clarification in the final version that the proposed back-patching method is a proof-of-concept. Finally, reviewer KJ8j had concerns that the baseline performance on real-world tasks are much lower than publicly reported results, likely due to the implementation / prompting design choices made to facilitate the analysis.

Overall, the submission makes valuable contributions to the multimodal learning and mechinterp communities, and the AC believes it is ready to be shared with the NeurIPS audience, assuming the promised clarifications will be made in the camera ready version. The AC also shares concerns (KJ8j, 1QgH) on the experimental setup and the generalizability of the empirical observations, but does understand novel methods and insights with a proof-of-concept implementation would still be impactful for the relevant research communities.